# Logic of Hypotheses: from Zero to Full Knowledge in Neurosymbolic Integration

## Abstract

Neurosymbolic integration (NeSy) blends neural-network learning with symbolic reasoning. The field can be split between methods injecting hand-crafted rules into neural models, and methods inducing symbolic rules from data. We introduce *Logic of Hypotheses* (LoH), a novel language that unifies these strands, enabling the flexible integration of data-driven rule learning with symbolic priors and expert knowledge. LoH extends propositional logic syntax with a *choice operator*, which has learnable parameters and selects a subformula from a pool of options. Using fuzzy logic, formulas in LoH can be directly compiled into a differentiable computational graph, so the optimal choices can be learned via backpropagation. This framework subsumes some existing NeSy models, while adding the possibility of arbitrary degrees of knowledge specification. Moreover, the use of Gödel fuzzy logic and the recently developed Gödel trick yields models that can be discretized to hard Boolean-valued functions without any loss in performance. We provide experimental analysis on such models, showing strong results on tabular data and on the Visual Tic-Tac-Toe NeSy task, while producing interpretable decision rules.

## 1 Introduction

Neurosymbolic integration (NeSy) tries to combine the symbolic and sub-symbolic paradigms. The aim is to retain the clarity and deductive power of logic while leveraging the learning capabilities of neural networks (Besold et al., 2021; Marra et al., 2024). In many NeSy approaches, such as DeepProbLog (Manhaeve et al., 2018) and LTN (Badreddine et al., 2022), domain experts provide prior knowledge in the form of logic formulas, which the neural model uses as a bias or as constraints. While this strategy has proven effective, it presupposes that high-quality rules are readily available. On the other hand, other NeSy methods have approached the learning of symbolic knowledge from data in the field of rule mining (Qiao et al., 2021; Katzir et al., 2020; Wang et al., 2020). Further, some advanced methods can simultaneously learn symbolic rules and perception-to-symbol mappings (Wang et al., 2019; Daniele et al., 2022; Barbiero et al., 2023), thereby grounding symbols to raw data while simultaneously discovering the logical structure.

These research threads occupy opposite ends of a spectrum: knowledge-*injection* versus rule-*induction*. However, they typically lack the flexibility to handle intermediate scenarios in which a *partial logical structure* is supplied and the missing parts must still be learned. Such situations arise, for example, when existing prior knowledge must be revised or completed, or when learned rules are required to respect specific syntactic templates (e.g., CNF, DNF, Horn clauses, fixed-length clauses) for which the model was never programmed. Addressing this flexible middle ground remains an open challenge for NeSy methods.

In this paper, we propose the *Logic of Hypotheses* (LoH), a new logical formalism introducing the *choice operator*, which learns to select a subformula from a set of candidates. Such language can be used to produce neural networks with varying degree of symbolic bias, ranging from complete prior knowledge to none. In this way, LoH offers a single, principled learning framework that can adapt to the amount and form of prior knowledge available. Our main contributions are:

- **A novel language (LoH)** that extends propositional logic syntax with a *choice operator*, making it possible to leave parts of a formula underspecified. This allows to represent a hypothesis space $\mathcal{H}$ of formulas, in a flexible and compact way.

- **A compilation procedure** producing a differentiable computational graph from any LoH formula Φ, allowing for the learning of a data-fitting logical formula among those in the hypothesis space represented by Φ. We employ the Gödel trick (Daniele & van Krieken, 2025), a newly proposed stochastic variant of Gödel logic. Thanks to this choice, our approach can directly learn discrete functions through backpropagation. Moreover, the computational graph can be stacked on top of a neural network, allowing the end-to-end learning of symbolic rules alongside perception-to-symbol mappings.

- **A unifying viewpoint** of NeSy paradigms. The general neural layers of rule-inducing NeSy models like Wang et al. (2020); Payani & Fekri (2019) can be seen as the compilation of particular LoH formulas using product fuzzy logic. On the other hand, the full injection of prior knowledge can be obtained by simply avoiding the usage of the choice operator in a LoH formula. However, LoH is not limited to those two extremes and allows to construct models for many different intermediate situations (see Section 6).

## 2 RELATED WORKS

**Logics on top of Neural Predicates.** Approaches such as *SBR* (Diligenti et al., 2017), *LTN* (Badreddine et al., 2022) and Semantic Loss (Xu et al., 2018) translate logical knowledge into differentiable penalties added to the loss. In contrast, abductive methods (Dai et al., 2019; Tsamoura et al., 2021; Huang et al., 2021) let a symbolic model find labels for the neural part consistent with the provided knowledge. This enables logical reasoning also at inference time, yet the symbolic program remains user-supplied rather than learned. Similarly, *DeepProbLog* (Manhaeve et al., 2018), *DeepStochLog* (Winters et al., 2022), and *NeurASP* (Yang et al., 2020), enrich logical solvers with neural predicates whose outputs are treated as probabilities. The *Gödel Trick* (Daniele & van Krieken, 2025) makes formulas differentiable via Gödel semantics, and add noise to avoid local minima. Optimizing with backpropagation can then be interpreted as a (discrete) local search algorithm for SAT solving. On the rule-inducing side, *SATNet* (Wang et al., 2019) embeds a smoothed MaxSAT layer inside a neural network, jointly optimizing clause weights and perception. Subsequent work exposed limitations with unsupervised grounding (Chang et al., 2020), partially alleviated in Topan et al. (2021). *DSL* (Daniele et al., 2022) directly learn symbolic rules from data alongside the perception-to-symbol mappings, but the symbolic part is only a lookup table.

**Neural Networks with Soft Gates.** Many recent NeSy learners devise neurons that perform a continuous relaxation of the AND and OR operations, with learnable weights acting as soft gates. These neurons are placed into layers alternating the two operations, while the NOT operation is obtained by doubling the inputs—juxtaposing each input value with its negation. Models like these are typically used to learn propositional rules on binarized tabular data (Qiao et al., 2021; Dierckx et al., 2023; Katzir et al., 2020; Kusters et al., 2022). Among these, *Multi-Layer Logical Perceptron* (MLLP) (Wang et al., 2020) is a state-of-the-art model, using product fuzzy logic operations. However, like all the others, it does not guarantee that the extracted rules—which it calls *Concept Rule Sets (CRS)*—have the same accuracy as the neural model. Instead, we propose to use Gödel fuzzy logic, which allows a lossless extraction. Soft logical gates are also employed on binary/ternary neural networks (Deng et al., 2018), which are typically used for the efficiency of the quantized networks at inference, rather than the logical semantics. In particular, *Differentiable Logic Networks* (DLN) (Petersen et al., 2022; 2024) keep a sparse fixed wiring with nodes having at most two parents, and learn which binary Boolean operation each node should execute. Instead, LoH does the opposite: the logical operations are fixed, and the learnable gates decide which branch is selected. This allows LoH to yield more readable formulas (as DLNs rely on deeply nested structures employing 16 different logical operators), and offers a more straightforward path for incorporating prior knowledge.

**Inductive Logic Programming (ILP).** Also modern NeSy methods in ILP, such as *dILP* (Evans & Grefenstette, 2018), *NTP* (Campero et al., 2018) and *NeurRL* (Gao et al., 2025), use neurons performing a soft version of the logical operations, thus learning first-order rules via gradient descent. Of particular interest are *Logical Neural Networks* (LNNs) (Riegel et al., 2020), which compile formulas into neural networks with weighted Łukasiewicz operators, but rely on constrained optimization (e.g., Frank-Wolfe (Frank et al., 1956)), which hampers scalability. Moreover, assigning "importances" to the subformulas, instead of choosing one among the candidates as in LoH, limits the possibility to extract hard rules without loss in accuracy.

**Neuro-fuzzy networks (NFNs).** NFNs (e.g., *ANFIS* (Jang, 1993), *LazyPOP* (Zhou & Quek, 1996) and *GSETSK* (Nguyen et al., 2015)) are interpretable models where a neural network structure directly coincides with a fuzzy rule base. Typically they concentrate on parametric identification under fixed rules. Even when logic is induced (Shihabudheen & Pillai, 2018), their rule-based architecture is suited for a DNF-like format, which contrast the expressivity and flexibility of LoH.

## 3 BACKGROUND

Classical propositional logic builds formulas from propositional variables using negation ($\neg$), conjunction ($\wedge$) and disjunction ($\vee$).[1] An interpretation assigns to each variable the Boolean value true (1) or false (0), and extends recursively: the interpretation of $\neg\phi$ flips the value of $\phi$, the one of $\phi\wedge\psi$ returns the conjunction (AND) of the two values, and $\phi\vee\psi$ returns the disjunction (OR). Fuzzy logics relax the interpretations' truth values to the real unit interval $[0,1]$, interpreting connectives with *t-norms* (for $\wedge$) and *t-conorms* (for $\vee$). This relaxation brings differentiable operations, allowing gradient-based optimization. Common fuzzy logics include Łukasiewicz, Product, and Gödel. Product logic has $t(x,y) := xy$ as t-norm (i.e., conjunction), and $s(x,y) := 1-(1-x)(1-y) = x+y-xy$ as t-conorm (i.e., disjunction). On the other hand, Gödel logic uses $\min$ and $\max$ for conjunction and disjunction, respectively. In both, the negation of $x$ corresponds to $1-x$.

Gödel logic stands out for its simplicity and its closer alignment to classical logic in terms of idempotency and distributivity. Importantly, Gödel logic has the following property:

**Theorem 1** (Theorem 4.1 in Daniele & van Krieken (2025))**.** *For any Gödel interpretation $\mathcal{G}$, let $\mathcal{B}$ be the Boolean interpretation obtained rounding every fuzzy value in $\mathcal{G}$ with the thresholding function*[2]

$$\rho\colon [0,1] \setminus \{0.5\} \to \{0,1\}, \quad x \mapsto \begin{cases} 1 & \text{if } x > 0.5 \\ 0 & \text{if } x < 0.5 \end{cases} \tag{1}$$

*meaning that $\mathcal{B}(v_i) = \rho(\mathcal{G}(v_i))$ for every propositional variable $v_i$. Then, $\mathcal{B}$ is always consistent with $\mathcal{G}$, i.e., $\mathcal{B}(\phi) = \rho(\mathcal{G}(\phi))$ for every formula $\phi$.*

We can think at any logical formula as our model, with the truth values of the variables as inputs and the corresponding truth value of the formula as output. It is continuous if using fuzzy interpretations, and discrete if using classical Boolean ones. The theorem means that discretizing the outputs of the continuous model is the same as working with the discrete one, on the discretized inputs.

The main problem of using Gödel semantics is that its optimization can stall in shallow local minima. The Gödel Trick (Daniele & van Krieken, 2025) counters this by adding noise to each parameter, turning the optimization into a stochastic local search that escapes plateaus while remaining gradient-based. In practice, it works by storing as parameters the logits of the fuzzy weights. Then, for every step of training, random noise is sampled and added to the logits. This is done in the forward pass, before applying the sigmoid function producing the fuzzy weights. For a more complete discussion, see Appendix A or Daniele & van Krieken (2025).

## 4 LOGIC OF HYPOTHESES (LOH)

We introduce *Logic of Hypotheses* (LoH) first as a language for expressing hypothesis spaces (i.e., sets) of formulas in compact way. Syntactically, LoH extends propositional logic, adding a new *choice operator* $[\cdot]$ that can take as input any finite number of formulas:

$$F ::= \top \;\mid\; \bot \;\mid\; v \;\mid\; \neg F_1 \;\mid\; F_1 \vee F_2 \;\mid\; F_1 \wedge F_2 \;\mid\; [F_1, F_2, \ldots, F_n]$$

where $n$ can vary among the positive integers and $v \in \mathcal{V}$ represents the propositional variables. Semantically, a LoH formula $\Phi$ represents an entire set of classical propositional formulas $\mathcal{H}(\Phi)$, each obtained by selecting exactly one subformula per choice operator.

---

[1]For simplicity, we do not consider implication ($\rightarrow$) and iff ($\leftrightarrow$). However, there is nothing preventing their use, if associated with a consistent fuzzy semantics. For example, we may consider *material* implication, which comes from substituting $\phi \rightarrow \psi$ with its Boolean equivalent $\neg\phi \vee \psi$.

[2]The exclusion of $x = 0.5$, where negation would break the homomorphism property, is mostly a technicality. In practical settings, this exact value has measure zero in continuous-valued interpretations and does not affect the general applicability of the result.

**Example 1.** *The LoH formula* $\Phi := [a, b] \wedge [c, d] \wedge \neg e$ *has hypothesis space*

$$\mathcal{H}(\Phi) := \{ a \wedge c \wedge \neg e, \, a \wedge d \wedge \neg e, \, b \wedge c \wedge \neg e, \, b \wedge d \wedge \neg e \}$$

In general, the set $\mathcal{H}(\Phi)$ is obtained by applying inductively the following operations:

R1:  $\mathcal{H}(v) = \{v\}$; $\mathcal{H}(\top) = \{\top\}$; $\mathcal{H}(\bot) = \{\bot\}$;

R2:  $\mathcal{H}(\neg F_1) = \{\neg \phi \mid \phi \in \mathcal{H}(F_1)\}$;

R3:  $\mathcal{H}(F_1 \wedge F_2) = \{\phi \wedge \psi \mid \phi \in \mathcal{H}(F_1), \, \psi \in \mathcal{H}(F_2)\}$;

R4:  $\mathcal{H}(F_1 \vee F_2) = \{\phi \vee \psi \mid \phi \in \mathcal{H}(F_1), \, \psi \in \mathcal{H}(F_2)\}$;

R5:  $\mathcal{H}([F_1, \ldots, F_n]) = \bigcup_{i=1}^{n} \mathcal{H}(F_i)$.

**Example 2.** *Let's unfold the step-by-step procedure for producing* $\mathcal{H}([a, [b, c]] \wedge \neg [c, d])$:

*(R1)*   $\mathcal{H}(a) = \{a\}$;   $\mathcal{H}(b) = \{b\}$;   $\mathcal{H}(c) = \{c\}$;   $\mathcal{H}(d) = \{d\}$;

*(R5)*   $\mathcal{H}([b, c]) = \{b, c\}$;   $\mathcal{H}([c, d]) = \{c, d\}$;

*(R5)+(R2)*   $\mathcal{H}([a, [b, c]]) = \{a, b, c\}$;   $\mathcal{H}(\neg [c, d]) = \{\neg c, \neg d\}$;

*(R3)*   $\mathcal{H}([a, [b, c]] \wedge \neg [c, d]) = \{a \wedge \neg c, a \wedge \neg d, b \wedge \neg c, b \wedge \neg d, c \wedge \neg c, c \wedge \neg d\}$

In neural networks, the output of a hidden neuron is usually fed to multiple neurons of the subsequent layer. Similarly, in LoH, we may want to use the same "choice" of a subformula in multiple places. This can be solved by defining a placeholder for a sub-formula, and use it in multiple parts of the main formula. The algorithm for producing the hypothesis space corresponding to an LoH formula with such placeholders is reported in Appendix B.

**Example 3.** *The LoH formula* $[a, b] \wedge [a, b]$ *has hypothesis space* $\{a \wedge a, a \wedge b, b \wedge a, b \wedge b\} \equiv \{a, a \wedge b, b\}$. *On the other hand, the LoH formula* $\phi \wedge \phi$ *with* $\phi := [a, b]$ *has hypothesis space* $\{a \wedge a, b \wedge b\} \equiv \{a, b\}$. *In* $[a, b] \wedge [a, b]$ *the two choices are independent, whereas* $\phi \wedge \phi$ *introduces just one choice and reuses the selected subformula twice.*

Notice that LoH is flexible enough to encode *any* finite set of propositional formulas $\{h_1, \ldots, h_n\}$. Indeed, $[h_1, \ldots, h_n]$ represents exactly that space, even if more compact representations—whose compilations will require less parameters—may be possible. Even for a fixed hypothesis space, LoH is flexible enough to provide formulas biasing the search process in different ways. For example, both $[a, [b, c]]$ and $[a, a, b, c]$ are more biased towards choosing $a$ than $[a, b, c]$.

## 5   FROM LoH TO DIFFERENTIABLE COMPUTATIONAL GRAPHS

In the preceding section, we presented LoH as a language for expressing hypothesis spaces of logical formulas. We now show how the same LoH formulas can be turned into supervised machine learning models searching in those hypothesis spaces. The search will be done by gradient descent with backpropagation, so we need to compile LoH formulas into differentiable computational graphs.

The first step is to introduce a weight $w_i \in [0, 1]$ for every candidate subformula $F_i$ inside a choose operator. These are learnable and act as gates. Each choice operator is then converted to a propositional formula linking such weights to the respective subformulas. This can be done in two dual, practically interchangeable ways, which we call *disjunctive/conjunctive compilations*:

*Disjunctive Compilation*      $[F_1, \ldots, F_n] \quad \rightsquigarrow \quad \bigvee_{i=1}^{n} w_i \wedge F_i$

*Conjunctive Compilation*      $[F_1, \ldots, F_n] \quad \rightsquigarrow \quad \bigwedge_{i=1}^{n} \neg w_i \vee F_i$

Whichever of the two we use—more on this later—, we are left with a propositional formula with only the operators $\neg$, $\wedge$ and $\vee$. In order to have differentiable operations, we interpret them under a fuzzy

semantics. For example, with Gödel fuzzy logic, $\bigvee_{i=1}^n w_i \wedge F_i$ becomes $\max_{i=1,\ldots,n}(\min(w_i, F_i))$ and $\bigwedge_{i=1}^n \neg w_i \vee F_i$ becomes $\min_{i=1,\ldots,n}(\max(1 - w_i, F_i))$. The final piece is to design the weights in such a way that they can take continuous values in the interval $[0, 1]$, while allowing to extract the discrete selection of a candidate subformula for every choice operator.

**Design of the weights.** Let us first consider the case in which the weights are binary, i.e., each $w_i$ can only have value 0 or 1. Then, the formulas above can be simplified to equivalent ones, recalling that $0 \wedge F \equiv 0$, $1 \wedge F \equiv F$, $0 \vee F \equiv F$ and $1 \vee F \equiv 1$, for any formula $F$. It follows that the disjunctive (resp. conjunctive) compilation is equivalent to the disjunction (resp. conjunction) of the subformulas with weight 1. So if we impose that one weight $w_i$ is 1 and the remaining are 0, then both the conjunctive and the disjunctive compilation become equivalent to the single "chosen" subformula—the one with $w_i = 1$. This is exactly the condition we want after discretizing the weights. Indeed, we want the discrete selection of a *single* candidate from each choice operator.

The simplest way to discretize the weights is to use the thresholding function $\rho$ defined in equation 1. Hence, for any tuple of weights $(w_1, \ldots, w_n) \in [0, 1]^n$ associated to a choice operator $[F_1, \ldots, F_n]$, we want to impose that $w_i > 0.5$ for one and only one $i$. Instead of storing the weights $w_i$ directly, let us associate to each of them the actual learnable parameter $z_i$, which can take any real value. The differentiable operations for deriving the weights $w_i$ from these parameters are the following:

1. To escape local minima in the optimization procedure, at each *forward step of training*, add random noise to the parameters: $z_i' := z_i + n_i$ with $n_i \sim Gumbel(0, \beta)$ and $\beta$ being an hyperparameter.

2. Let $\bar{z}'$ be the mean of the two largest $z_i'$ values, and subtract it to each $z_i'$. By construction, all points $z_i'$ but the largest lie on the left of $\bar{z}'$.[3] Hence, one and only one among the shifted values $z_i'' := z_i' - \bar{z}'$ is positive.

3. Apply the sigmoid function: $w_i := \sigma(z_i''/T) = \frac{\exp(z_i''/T)}{1+\exp(z_i''/T)}$, with the temperature $T$ being an hyperparameter.

The application of the sigmoid function ensures that the weights are in the interval $[0, 1]$. Moreover, because of the second step, one and only one logit $z_i''$ is positive, so exactly one weight $w_i$ is greater than 0.5. This procedure for producing weights $w_i$ from the $z_i$'s is an adaptation of the Gödel trick with categorical variables (Daniele & van Krieken, 2025), and is further discussed in Appendix A.

**Disjunctive vs Conjunctive Compilation.** Both have the same purpose of selecting *one* subformula among the candidates, and in general both work well. We suggest using the former when the choice operator is a term in a disjunction, and the latter when in a conjunction; the opposite if negated. For example, for $\neg[a, b] \wedge [b, c] \wedge ([c, d] \vee \neg[d, e])$, we suggest using disjunctive compilation for $[a, b]$ and $[c, d]$, and conjunctive compilation for $[b, c]$ and $[d, e]$. The theoretical and empirical motivations for this are discussed in Appendix C.

**Robustness to Binarization.** By design, it is always possible to binarize the weights of a model and extract the chosen subformula from each choice operator. The result is a propositional formula in the hypothesis space denoted by the LoH formula. If using Gödel fuzzy logic, then Theorem 1 guarantees that the outputs of the resulting propositional formula *coincide* with the rounding of the outputs of the continuous model on *every* possible input.[4] Accuracy, confusion matrices, and any higher-level deductive tasks are thus preserved from the continuous to the discrete model. In this sense, Gödel logic offers lossless rule extraction. Notice that this is true also at any point in training. So the entire training process—while being gradient-based—can be interpreted symbolically, through discrete changes. To our knowledge, no other rule-learning NeSy model achieves this.

---

[3]The probability of the two largest values coinciding is negligible, especially after adding the continuous Gumbel noise. Moreover, this happening would affect only the extraction of hard rules, not the optimization procedure (which would continue to change the parameters, eventually breaking the tie).

[4]See Appendix D for a counterexample with product fuzzy logic.

## 6 LoH as a Unifying Framework for NeSy Integration

In this section, we demonstrate how LoH captures a wide range of existing NeSy methods—spanning fully-provided prior knowledge, partially known templates, and purely data-driven rule induction. To illustrate these settings, we instantiate them on the same small synthetic task: a wildfire–risk assessment problem.

**Experimental Setup: Wildfire Risk.** We generated a dataset of 2048 samples, each consisting of a simulated aerial image and a set of 7 Boolean environmental features (e.g., $StrongWind$, $LowHumidity$). The task is to predict a binary $WildfireRisk$ label. The model architecture consists of a generic CNN $h_\theta$ that processes the image to predict two latent concepts, $DenseForest$ and $DryVegetation$, and a logical component that combines these visual concepts with the environmental features to produce the final prediction. Crucially, the CNN is not pretrained. It must learn to identify forests and vegetation solely via the gradients backpropagated through the logic. The ground-truth label is generated by the conjunction of three rules:

$$
\begin{align}
Fuel &:= DenseForest \vee (DryVegetation \wedge StrongWind) \tag{2a} \\
DryConditions &:= LowHumidity \vee (HighTemperature \wedge \neg RainedRecently) \tag{2b} \\
Trigger &:= LightningsFrequent \vee \neg LowHumanActivity \vee PowerLinesNearby \tag{2c} \\
WildfireRisk &:= Fuel \wedge DryConditions \wedge Trigger \tag{2d}
\end{align}
$$

We analyze how LoH handles this task under different knowledge assumptions. See Appendix E for full implementation details.

**Full Knowledge (No Choice Operators).** If the choice operator $[\cdot]$ is never used, the logical part has no learnable parameter, but the gradient can backpropagate from it to a neural part below. This corresponds to many well-known NeSy approaches where knowledge is entirely specified a priori.

In the wildfire experiment, we set the LoH formula exactly to the ground-truth rule (2d). Since the logic is fixed and correct, the learning focuses entirely on the perception module. The model successfully solves this symbol grounding problem, learning to map images to $DenseForest$ and $DryVegetation$ with high accuracy simply by minimizing the classification error of the $WildfireRisk$ label.

**Selecting Reliable Rules.** Suppose we have a knowledge set of $n$ candidate rules $r_1, \ldots, r_n$, but we are unsure on whether they are correct. Then, we can use the following LoH formula to select a reliable subset:

$$
\bigwedge_{i=1}^{n} [r_i, \top] \tag{3}
$$

Indeed, the hypothesis space spans all possible subsets: by selecting $r_i$ over $\top$, the model effectively picks such rule. When the rules are clauses (i.e., disjunctions of possibly negated propositions), this setup parallels *KENN* (Daniele & Serafini, 2019), which also have a learnable weight for each rule (even if it is never made discrete). In our framework, each $r_i$ in equation 3 can be any formula. For example, we may have entire knowledge bases as $r_i$'s, and use equation 3 to decide which to trust.

On the wildfire task, we build such a pool by taking the three ground-truth rules $Fuel$, $DryConditions$, and $Trigger$ and augmenting them with plausible but incorrect variants. Formula 3 is then applied to this pool. The resulting learned subset is a sparse, data-driven refinement of the original possibilities.

**Selecting One Rule per Set.** Given $m$ sets $\{r_{i,1}, \ldots, r_{i,n_i}\}$ of candidate rules, we may want to enforce the choice of *exactly one* rule per set. This may happen for example because the rules in each set are mutually exclusive. For this purpose, we can use the following LoH formula:

$$
\bigwedge_{i=1}^{m} [r_{i,1}, r_{i,2}, \ldots, r_{i,n_i}] \tag{4}
$$

For wildfire risk, suppose domain experts agree on the structure of the final rule as a conjunction of three components—fuel, dryness, trigger—but propose several alternative for each component. We group these into three sets and apply equation 4 with $m = 3$: one choice among fuel candidates, one among dryness candidates, and one among trigger candidates. This regime differs from the previous one in that the model must commit to a single alternative within each group, instead of freely activating or deactivating rules.

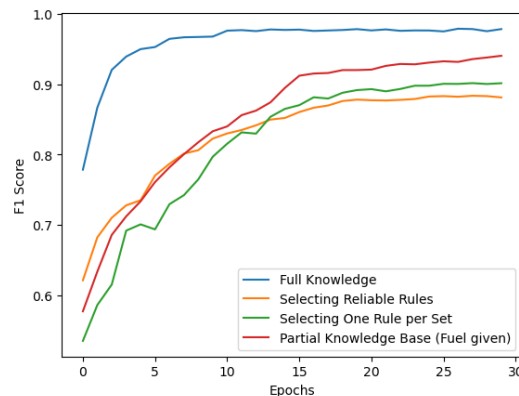

Figure 1: Average training curves, over 20 runs, of the LoH models based on different levels of knowledge, on the wildfire risk assessment task. Richer priors yield better performance.

**Partial Knowledge Base.** If we have a knowledge base $K$ which is reliable but not complete, we can couple it with a rule-learning LoH formula $\Phi$, and use $K \wedge \Phi$. Similarly, we may have a knowledge base whose rules have some missing parts, and fill them with some rule-learning formulas.

As an example, we consider the $Fuel$ rule as given, while the rest is unknown and must be selected from the same pool of rules used in the previous settings. For these candidates, we apply the subset-selection scheme of equation 3, so that LoH learns which dryness- and trigger-related rules best complement the known $Fuel$ rule. Figure 1 compares this and the other knowledge regimes introduced so far.

**Zero Knowledge (Pure Rule Learning).** For rule induction, we can combine neurons learning disjunctions and neurons learning conjunctions. A simple and most efficient way is to arrange them in layers and exploit parallel computation on tensors, like in standard Artificial Neural Networks. The following are LoH formulas for neurons learning the disjunction of a subset of neurons of the previous layer, with and without negations:

$$n_j^{(l+1)} := \bigvee_{i=1}^{m_l} [n_i^{(l)}, \neg n_i^{(l)}, \bot] \qquad \text{and} \qquad n_j^{(l+1)} := \bigvee_{i=1}^{m_l} [n_i^{(l)}, \bot] \qquad (5)$$

Analogously, conjunctive neurons simply replace disjunction with conjunction and False with True. The layers adopted in NLNs (Payani & Fekri, 2019) and MLLPs (Wang et al., 2020) use neurons analogous to those, with product fuzzy logic, instead of Gödel's. In this correspondence, they would use—as we suggested—conjunctive compilation for the choice operators in conjunctive neurons and vice versa for disjunctive neurons. However, LoH is flexible and many alternative designs of neurons are possible. For example,

$$n_j^{(l+1)} := \bigvee_{i=1}^{k} [n_1^{(l)}, \ldots, n_{m_l}^{(l)}, \neg n_1^{(l)}, \ldots, \neg n_{m_l}^{(l)}] \quad \text{and} \quad n_j^{(l+1)} := \bigvee_{i=1}^{k} [n_1^{(l)}, \ldots, n_{m_l}^{(l)}] \qquad (6)$$

are neurons learning clauses of fixed width $k$ (possibly with repetitions)—$k$ being a hyperparameter. A detailed treatment of this zero-knowledge regime is deferred to the next section.

**Respecting Syntactic Requirements.** If the learned rules of a NeSy model are to be used also in a symbolic program, this may require them to adhere to a specific format or template. If it is possible to express the template in LoH—and LoH is flexible in this regard—, then the adherence is guaranteed. As an example, here is a possible template for clauses of width 3:

$$[v_1, \ldots, v_n, \neg v_1, \ldots, \neg v_n] \ \vee \ [v_1, \ldots, v_n, \neg v_1, \ldots, \neg v_n] \ \vee \ [v_1, \ldots, v_n, \neg v_1, \ldots, \neg v_n] \qquad (7)$$

And here is a template for definite clauses (i.e., clauses with exactly one positive variable):

$$\bigvee_{i=1}^{n} [\neg v_i, \bot] \ \vee \ [v_1, \ldots, v_n] \qquad (8)$$

Experiments on enforcing these templates are provided in Appendix F. We employ a distinct, purely symbolic dataset to better highlight the model's convergence properties for different syntactic requirements.

# 7 EXPERIMENTS

In this section, we evaluate the performance of our models, in their general form when no domain knowledge is provided: i.e., models made alternating conjunctive and disjunctive layers, with neurons of the form in equation 5 or in equation 6. We employ the Gödel trick and test the performance on several benchmark classification datasets. All experiments were conducted on a cluster node equipped with an Nvidia RTX A5000 with 60GB RAM.

## 7.1 CLASSIFICATION PERFORMANCE ON TABULAR DATASETS

**Datasets.**    We use 12 classification benchmarks from the UCI Machine Learning Repository, available with CC-BY 4.0 license. They were previously employed in Wang et al. (2020) for evaluating MLLP. As a preprocessing step, we adopt the same data discretization and binarization procedure as in Wang et al. (2020): the recursive minimal entropy partitioning algorithm (Dougherty et al., 1997), followed by one-hot-encoding. Datasets' references and properties are available in Appendix H.

**Models.**    In our model, we alternate conjunctive and disjunctive layers without negations. The choice between using neurons of type (5) or neurons of type (6) is performed by the hyperparameters' tuning, together with the rest of the architecture (number and size of layers, and whether to start with a conjunctive or a disjunctive layer). We compare our model against Differentiable Logic Networks (DLN) (Petersen et al., 2022), the aforementioned MLLP (Wang et al., 2020), and the rules extracted from it, which are called CRS. Note that for DLN we report the performance of the continuous model, which is generally higher than the discretized one. Instead, Gödel semantics ensures our model behaves identically before and after binarization, and CRS corresponds to the post-hoc binarization of MLLP. We also consider commonly used machine learning baselines: Decision Trees (DT), Random Forests (RF), XGBoost and standard fully-connected Neural Networks (NN).

Five of the twelve datasets have more than two classes, and we want to treat multi-class predictions as mutually exclusive output propositions. Therefore, in our model, we apply a reparameterization to the final layer, to guarantee that exactly one output per example exceeds the threshold value $0.5$—the output of the predicted class. This is analogous to the procedure used before for designing the weights of a choice operator, but does not require the addition of noise. Concretely, let $z_i$ be the logits of the outputs $o_i$ produced by the outmost layer—each one corresponding to a different class. Each logit gets shifted by subtracting the mean $\bar{z}$ of the two largest $z_i$'s. The resulting $z_i - \bar{z}$ values—of which, by construction, one and only one is positive—are then outputted through a sigmoid activation.

**Methodology.**    Each dataset is divided into $20\%$ for testing and $80\%$ for training and validation. In particular, validation takes $12.5\%$ of the second split, so $10\%$ of the whole dataset. Since most datasets are unbalanced, we use the F1 score (macro) as classification metric. As loss functions, we use Binary Cross-Entropy for NN, DLN and our model, and Mean Squared Error for MLLP. All of them are optimized using Adam (Kingma, 2014). For the selected hyperparameters, Table 1 provides the means and standard deviations of the test-set F1 scores, out of 10 different training runs on the training plus validation sets. For each dataset, the hyperparameters of each model are tuned using 80 trials of the TPE algorithm (Bergstra et al., 2011) from the Optuna library. Architectural choices—such as the number and width of layers—are tuned alongside all other hyperparameters. Appendix I lists every tuned variable (and their search ranges), while the values ultimately selected are available in the code.

**Discussion.**    Although vanilla neural networks attain the best summary score (.89) and rank (2.9), they—together with RF and XGBoost—do not allow to extract symbolic knowledge. MLLP—which come second—does support rule extraction, but with the loss in performance from it to CRS. In fact, only DT, CRS, the discretization of DLN, and our model achieve all the benefits of symbolic discrete rules, such as better interpretability, explainability, and efficiency of inference on CPUs. Moreover, unlike DT, RF and XGBoost, our model is fully differentiable, so it could be placed downstream of perception modules (CNNs, transformers, etc.) and be trained end-to-end.

Apart for the two datasets with the largest numbers of classes, namely *chess* (with 18) and *letRecog* (with 26), our model have consistently better scores than CRS, and is almost always on par with, or close to, the neural network baseline. This suggests that our handling of many-way classification

Table 1: Mean F1 scores on the test sets of tabular benchmarks, out of 10 runs.

| Dataset | DT | RF | XGBoost | NN | DLN | MLLP | CRS | Ours |
|---|---|---|---|---|---|---|---|---|
| adult | .80 ±.00 | .81 ±.00 | **.82** ±.00 | .81 ±.01 | .80 ±.02 | .80 ±.01 | .75 ±.06 | .81 ±.01 |
| bank-m. | .74 ±.00 | .73 ±.00 | .76 ±.00 | **.78** ±.01 | 76 ±.01 | .69 ±.08 | .75 ±.01 | .76 ±.01 |
| banknote | .95 ±.00 | .95 ±.01 | **.96** ±.00 | **.96** ±.00 | .95 ±.01 | **.96** ±.00 | **.96** ±.00 | **.96** ±.00 |
| blogger | .64 ±.00 | .53 ±.00 | .69 ±.00 | .82 ±.05 | .72 ±.12 | **.84** ±.00 | .78 ±.02 | .83 ±.08 |
| chess | .81 ±.00 | .58 ±.01 | **.85** ±.00 | .82 ±.01 | .41 ±.02 | .83 ±.02 | .74 ±.02 | .69 ±.01 |
| connect-4 | .59 ±.00 | .55 ±.00 | **.71** ±.00 | **.71** ±.04 | .62 ±.01 | .58 ±.01 | .58 ±.01 | .58 ±.01 |
| letRecog | .80 ±.00 | .76 ±.00 | .92 ±.00 | **.93** ±.00 | .64 ±.02 | .85 ±.01 | .80 ±.01 | .77 ±.01 |
| magic04 | .81 ±.00 | .83 ±.00 | **.84** ±.00 | **.84** ±.00 | .83 ±.00 | **.84** ±.00 | .80 ±.00 | .83 ±.00 |
| mushroom | **1.** ±.00 | **1.** ±.00 | **1.** ±.00 | **1.** ±.00 | **1.** ±.00 | **1.** ±.00 | **1.** ±.01 | **1.** ±.00 |
| nursery | .79 ±.00 | .79 ±.00 | .80 ±.00 | **1.** ±.00 | **1.** ±.00 | **1.** ±.00 | **1.** ±.00 | **1.** ±.00 |
| tic-tac-toe | .89 ±.00 | .98 ±.01 | .97 ±.00 | .99 ±.01 | .99 ±.01 | **1.** ±.00 | **1.** ±.00 | .99 ±.01 |
| wine | **1.** ±.00 | **1.** ±.01 | .96 ±.00 | .97 ±.05 | .97 ±.02 | **1.** ±.00 | .96 ±.01 | .98 ±.01 |
| mean (↑) | .82 | .79 | .86 | **.89** | .81 | .87 | .84 | .85 |
| avg. rank (↓) | 5.7 | 6.0 | 3.7 | **2.9** | 5.2 | 3.5 | 5.0 | 4.1 |

may not be optimal. Notably, DLN struggles even more on those two benchmarks, and in general does not surpass our model even in its continuous (pre-discretization) form. Hence, when only a few classes are present, our model reliably ranks among the top performers while simultaneously providing symbolic rules and end-to-end differentiability.

## 7.2 VISUAL TIC-TAC-TOE

One of the previous classification benchmarks (Aha, 1991) is based on the game of tic-tac-toe. The dataset provides all possible ending board configurations (for games in which $X$ begins), and the target is to predict whether $X$ has won the game or not. So the target function can be expressed with a simple formula in Disjunctive Normal Form (DNF), having 8 clauses.[5]

Let us introduce a more challenging variant of the dataset, which we refer to as *Visual Tic-Tac-Toe*. In this version, instead of symbolic board encodings, each cell is represented by a MNIST image. Specifically, we assign images of the digit 0 to represent $X$, images of the digit 1 to represent $O$, and blank cells are represented by images of the digit 2. As a result, instead of a structured propositional encoding, the inputs consist of $3 \times 3$ grids of grayscale images. This modification significantly increases the difficulty of the task, as models must now learn to recognize digit representations from the images before they can reason about tic-tac-toe board configurations, but with as little supervision as before. Traditional symbolic models would struggle with such high-dimensional and noisy input, making this an interesting benchmark for neuro-symbolic learning.

**Models.** To handle the image-based input, we extend all models with a standard CNN. This network consists of two convolutional layers, each using a kernel size of 3 with padding 1, followed by ReLU activations and max-pooling. The first convolutional layer has 32 output channels, while the second has 64. After feature extraction, the CNN flattens the output and passes it through a fully connected layer of size 128, a dropout layer with probability 0.5, and another fully connected layer of size 3. The three outputs for each of the nine images composing a grid are then concatenated.

For DLN, MLLP/CRS and our models, the outputs of the CNN for the nine images are to be considered as input "propositions", so their values are clipped between 0 and 1. The learning process is end-to-end, including the CNN component. The hyperparameters (and their tuning) are as before, but we allow the CNN part to have a separate learning rate (in the range $10^{-5}$–$10^{-3}$). Moreover, for both MLLP/CRS and our model, we fix the number of layers to 2 and distinguish the two cases of whether the last one is disjunctive (DNF) or conjunctive (CNF). This is not possible for DLN.

---

[5]Let the input data be encoded with propositions $X_1$-$X_9$, $O_1$-$O_9$ and $B_1$-$B_9$, meaning that $X_i$ is true if there is an $X$ at position $(\lfloor i/3 \rfloor, i\%3)$, and similarly for $O$s and $B$lank cells. Then, the target function is $(X_1 \wedge X_2 \wedge X_3) \vee (X_4 \wedge X_5 \wedge X_6) \vee (X_7 \wedge X_8 \wedge X_9) \vee (X_1 \wedge X_4 \wedge X_7) \vee (X_2 \wedge X_5 \wedge X_8) \vee (X_3 \wedge X_6 \wedge X_9) \vee (X_1 \wedge X_5 \wedge X_9) \vee (X_3 \wedge X_5 \wedge X_7)$.

We also implemented a neural baseline that follows a classical deep learning approach (NN). This model uses a CNN module as the one described earlier, but its outputs are not clipped, and their number is a hyperparameter in the range 3-32 (instead of being fixed to 3). This is because in this case the outputs of this part of the network may be better seen as embeddings rather than symbols. The concatenation of such vectors (for the nine images in a grid) is then fed to a multi-layer perceptron.

**Methodology.** The original symbolic tic-tac-toe dataset is split with the same proportions as before (70%-10%-20%). The training and validation parts are given images from MNIST training set, while the test part is given images from MNIST test set. No image is used more than once. Moreover, for each board configuration in the training and validation sets, two different image grids are created, effectively doubling the number of samples.

Table 1 provides the mean and standard deviations of the test-set F1 scores, based on training each model 30 times on the combined training and validation sets. What changes between the runs is the network initialization, the mini-batches, the added noise in our model and the random binarization occurring in MLLP as a form of regularization. Beyond performance, a key advantage of NeSy approaches lies in their better interpretability. Since they operate on structured logical representations, we can analyze the learned decision rules after assigning semantic labels to the input units. The exact procedure for such labeling is reported in Appendix I, together with an example of the decision rules learned by each model. Table 1 also provides the average F1 scores of these extracted rules on the purely symbolic tic-tac-toe dataset.

Table 2: Comparison of the models on the Visual Tic-Tac-Toe task.

|  | NN | DLN | MLLP | | CRS | | Ours | |
|---|---|---|---|---|---|---|---|---|
|  |  |  | DNF | CNF | DNF | CNF | DNF | CNF |
| NeSy eval | .91 ±.14 | .96 ±.01 | .80 ±.22 | .94 ±.02 | .76 ±.28 | .92 ±.00 | **.97** ±.01 | .95 ±.01 |
| Symbolic eval | - | .37 ±.21 | - | - | .80 ±.30 | .97 ±.02 | **.99** ±.00 | **.99** ±.00 |

**Discussion.** On 2 out of its 30 training runs, the neural baseline remained stuck at always predicting the most frequent class, with macro F1 score of .39. Instead, on the other runs, its performance was comparable to that of our CNF model. Similarly, MLLP/CRS in the DNF setting remained stuck on 4 runs, with the remaining 26 performing slightly worse than our CNF model. Finally, MLLP in the CNF version, and DLN, have similar performance to our CNF model on all runs. However, the symbolic evaluation reveals that DLN fails when discretized, and MLLP/CRS learned less accurate symbolic rules than ours. Moreover, the DNF version of our model is consistently better than all other models, and also found the 100%-correct formula reported above on 4 occasions. These results highlight the ability of our models to recover symbolic decision rules with high fidelity, even when the symbols must be learned from continuous high-dimensional perceptions.

## 8 CONCLUSION AND FUTURE WORK

We introduced a single, compact language for expressing hypothesis spaces of logical formulas and compiling them into differentiable models whose discrete rule extraction is provably loss-free under Gödel semantics. LoH unifies knowledge injection and rule induction within one propositional paradigm, and yields strong results on both tabular and perceptual benchmarks, while retaining the possibility to extract learned logical formulas following arbitrary templates.

Despite these encouraging results, the present work leaves two important avenues open. First, the empirical validation should be broadened, targeting larger datasets, use of partial knowledge and additional NeSy tasks with complex perceptions. Second, the formalism is so far propositional. Extending LoH with first-order logic quantifiers would unlock relational reasoning and allow direct comparison with first-order NeSy learners. Addressing these two limitations—with richer logic and wider experimentation—constitutes our next research milestone.

## REPRODUCIBILITY STATEMENT

Assumptions, design choices and claims are reported in the main text and further explained in Appendices A and B. Theorem 1 is a generally known result, and a proof can be found in Daniele & van Krieken (2025). We share the code in the supplementary materials, and will make it public upon acceptance. Experimental setups—including dataset preprocessing, splits, metrics, protocol, etc.—are described in Section 7. Appendix I reports complete hyperparameter ranges, and the values taken for each benchmark are available within the code repository. Appendix H references the tabular datasets, and the code provide a way to build the Visual Tic-Tac-Toe benchmark. Finally, Appendix J explains the symbol-labeling procedure utilized in the Visual Tic-Tac-Toe experiment.

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

# A  GÖDEL TRICK

In this appendix, we provide a recap of the *Gödel trick with categorical variables*, adapting its original presentation in Daniele & van Krieken (2025, Appendix C) to our setting with fuzzy truth values in $[0, 1]$ discretized to $\{0, 1\}$ via a 0.5 threshold (instead of real values discretized to $\{-1, 1\}$ via the sign function). We also adapt the proof in Daniele & van Krieken (2025, Appendix D), showing that the Gödel Trick is equivalent to the classical *Gumbel-max trick* (Gumbel, 1954), of which the widely used *Gumbel-softmax trick* (Jang et al., 2017) is a smooth approximation.

**Gödel Trick Recap.**  Direct gradient descent on Gödel logic is prone to stalling in local minima. The Gödel Trick counteracts this by injecting noise. In our formulation, it takes a set of logits $z_i$ (one per candidate branch of a choice operator), and applies the following three steps, depicted in Figure 2:

1. **Noise addition (only during training):** $z_i' := z_i + n_i$, with $n_i \overset{\text{i.i.d.}}{\sim} \mathrm{Gumbel}(0, \beta)$.

2. **Re-centering:** $z_i'' := z_i' - \bar{z}'$, where $\bar{z}'$ is the mean of the two largest perturbed logits.

3. **Sigmoid application:** $w_i := \sigma\left(\frac{z_i''}{T}\right)$, where $T > 0$ is a temperature hyperparameter.

At test time, we can binarize the weights $w_i$ at 0.5, ensuring that exactly one $w_i$ equals 1 and the rest are 0. Thanks to Theorem 1, this discretization does not alter the final Boolean predictions of the model.

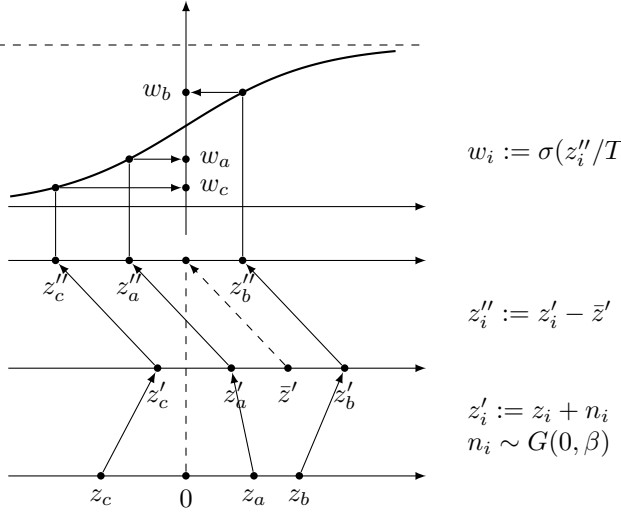

Figure 2: Given a choice operator $[a, b, c]$, the figure illustrates the differentiable three-stage procedure that turns the raw, real-valued logits $z_i$ attached to each candidate sub-formula into logical gates $w_i \in [0, 1]$. From the bottom up: (1) during training, i.i.d. Gumbel noise $n_i \sim \mathrm{Gumbel}(0, \beta)$ is added to each logit; (2) the mean $\bar{z}'$ of the two largest perturbed logits is subtracted from all values; (3) temperature-scaled sigmoid function is applied.

**Equivalence to the Gumbel-max Trick.**  The *Gumbel-max trick* is a reparameterization method for sampling from a categorical distribution. Let the categorical distribution be defined by the probabilities $\pi := softmax(\theta)$. Then, the probability that $\arg\max_i (\theta_i + g_i) = j$, where $g_i \sim Gumbel(0, 1)$, is precisely equal to $\pi_j$. Hence,

$$\mathrm{onehot}(\arg\max_i (\theta_i + g_i))$$

produces an exact sample from the categorical distribution with probabilities $\pi$. This is how the Gumbel-max trick works, and the Gumbel-softmax trick is a differentiable approximation which substitutes $\mathrm{onehot} \circ \arg\max$ with $softmax$.

Setting $\theta_i := z_i/\beta$ and $g_i := n_i/\beta$, the perturbation step of the Gödel Trick becomes:

$$z_i' = \beta\left(\theta_i + g_i\right)$$

The centering step subtracts the mean of the two largest perturbed logits, which does not affect the $\arg\max$, since it simply shifts all values by the same constant. Similarly, also the multiplication by a positive constant and the application of a monotonically increasing function such as the sigmoid do not affect the $\arg\max$. Hence:

$$\arg\max_i w_i = \arg\max_i z_i'' = \arg\max_i z_i' = \arg\max_i (\theta_i + g_i)$$

Since the maximum weight is by construction the only one surpassing the threshold value $0.5$, discretizing the weights $w_j$ yields precisely the same effect as $\mathrm{onehot}(\arg\max_i(w_i))$. Thus, we can conclude that

$$\rho(w_j) = \mathrm{onehot}(\arg\max_i (\theta_i + g_i))_j$$

for every $j$. Moreover, because we are using Gödel semantics and Theorem 1 applies, the thresholding of the weights is implicit when considering binarized output predictions of the entire model. Hence, in our context, the Gödel trick is equivalent to the Gumbel-max trick, which is more precise than the Gumbel-softmax one.

## B  LoH with Placeholders: a Formalization

**Placeholder Declarations.**    We allow the user to *name* any LoH subformula by writing a declaration of the form $p := \phi$, where $\phi$ is a LoH formula and $p$ is a fresh identifier (i.e., a name different from any propositional variable and any other identifier). A placeholder can itself mention other placeholders that have been declared earlier. However, the directed graph of such references must be acyclic, to avoid *circular* definitions.

Formally, let $\mathcal{P}l$ be the finite set of placeholder names and let $\mathcal{D} = \{p_1 := \phi_1, \ldots, p_m := \phi_m\}$ be the list of declarations. For every placeholder $p_i$, we define its *dependency set* $\mathrm{dep}(p_i) := \{p_j \in \mathcal{P}l \mid p_j$ occurs in $\phi_i\}$. The *dependency graph* is the directed graph $G = (\mathcal{P}, E)$ with edge $(p_i, p_j) \in E$ iff $p_j \in \mathrm{dep}(p_i)$. A declaration list is *well-defined* when $G$ is a directed acyclic graph (DAG).

**Hypothesis Space Computation.**    Let $\mathcal{D} = \{p_1 := \phi_1, \ldots, p_m := \phi_m\}$ be a well-defined list of declarations. The following algorithm produces the hypothesis space $\mathcal{H}(\Phi)$ for every LoH formula $\Phi$ possibly containing the placeholders $p_1, \ldots, p_n$.

---
**Algorithm 1** Construction of the hypothesis space $\mathcal{H}(\Phi)$

---
**Input:** LoH formula $\Phi$; well-defined declaration list $\mathcal{D} = \{p_i := \phi_i\}_{i=1}^m$
**Output:** $\mathcal{H}(\Phi)$, the hypothesis space of $\Phi$

1: **function** HYPOTHESES($\Phi, D$)
2:     **Tagging.** Traverse every choice node $[F_1, \ldots, F_n]$ in both $\Phi$ and the bodies $\phi_i$; assign a fresh identifier $c$ and record its arity $n_c$
3:     **Indices space.** $\mathsf{IndicesSpace} \leftarrow \prod_c \{1, \ldots, n_c\}$  (Cartesian product)
4:     **Evaluator.** Define the total function EVAL$(\cdot, \mathsf{Indices})$ recursively:

$$\mathrm{EVAL}(v, \mathsf{Indices}) := v \tag{R1'}$$
$$\mathrm{EVAL}(\neg F_1, \mathsf{Indices}) := \neg \mathrm{EVAL}(F_1, \mathsf{Indices}) \tag{R2'}$$
$$\mathrm{EVAL}(F_1 \wedge F_2, \mathsf{Indices}) := \mathrm{EVAL}(F_1, \mathsf{Indices}) \wedge \mathrm{EVAL}(F_2, \mathsf{Indices}) \tag{R3'}$$
$$\mathrm{EVAL}(F_1 \vee F_2, \mathsf{Indices}) := \mathrm{EVAL}(F_1, \mathsf{Indices}) \vee \mathrm{EVAL}(F_2, \mathsf{Indices}) \tag{R4'}$$
$$\mathrm{EVAL}([F_1, \ldots, F_n]_c, \mathsf{Indices}) := \mathrm{EVAL}(F_{\mathsf{Indices}[c]}, \mathsf{Indices}) \tag{R5'}$$
$$\mathrm{EVAL}(p_i, \mathsf{Indices}) := \mathrm{EVAL}(\phi_i, \mathsf{Indices}) \tag{Placeholders}$$

5:     **Enumeration.** $\mathcal{H}(\Phi) \leftarrow \{\mathrm{EVAL}(\Phi, \mathsf{Indices}) \mid \mathsf{Indices} \in \mathsf{IndicesSpace}\}$
6:     **return** $\mathcal{H}(\Phi)$
7: **end function**

---

Given $\Phi$ and $\mathcal{D}$, the Cartesian product $\mathsf{IndicesSpace}$ (in line 3) is exactly the set of all possible assignments of concrete choices to every choice operator. Indeed, for every $\mathsf{Indices} \in \mathsf{IndicesSpace}$,

the algorithm considers an index $\mathsf{Indices}[c] \in \{1, \ldots, n_c\}$ for every choice node $c$ (of arity $n_c$), and the evaluator $\mathrm{EVAL}(\cdot, \mathsf{Indices})$ is a function that (i) respects placeholder sharing and (ii) deterministically replaces every choice node $[F_1, \ldots, F_{n_c}]_c$ by the branch $F_{\mathsf{Indices}[c]}$. Thus, $\mathrm{EVAL}(\Phi, \mathsf{Indices})$ reflects the discrete model obtained compiling $\Phi$ (as in Section 5), when $\mathsf{Indices}[c]$ is the index of the unique weight greater than $0.5$, for each compiled choice operator $c$. It follows that $\mathcal{H}(\Phi)$ coincides exactly with the hypothesis space of the compiled and discretized model, as we wanted.

**Remark on DAG constraint.** Algorithm 1, as written in declarative pseudo-code, requires the declaration list $\mathcal{D} = \{p_i := \phi_i\}_{i=1}^m$ to be well-defined. In practice, however, our implementation does not enumerate the hypothesis space, and is written in PyTorch following a standard imperative framework. A placeholder's value is created the moment the corresponding tensor is defined, so a definition such as $\psi \leftarrow [\ldots, \phi, \ldots]$ must necessarily see $\phi$ already instantiated. And any subsequent attempt to define $\phi$ in terms of $\psi$ would simply create a new tensor rather than closing a cycle. Thus the execution order enforced by the host language ensures acyclicity "for free", making an explicit compile-time DAG check redundant. Importantly, this does not preclude recurrent structures: one can still build LoH-based recurrent networks exactly as one builds ordinary RNNs in PyTorch — by passing the hidden state from one time step to the next. In this case, one may write an LoH formula formally relating some placeholders to the ones at the previous time step, and acyclicity is in the unrolled architecture.

## C  DISJUNCTIVE VS CONJUNCTIVE COMPILATIONS

Recall that a choice node $[F_1, \ldots, F_n]$ can be compiled in two dual ways:

- Disjunctive compilation: $\bigvee_{i=1}^n (w_i \wedge F_i)$
- Conjunctive compilation: $\bigwedge_{i=1}^n (\neg w_i \vee F_i)$

Duality comes from De Morgan's laws: the negation of the disjunctive compilation is equivalent to the conjunctive compilation on the negated subformulas, and vice versa. Indeed,

$$\neg \bigvee_{i=1}^n w_i \wedge F_i \equiv \bigwedge_{i=1}^n \neg w_i \vee \neg F_i \qquad \text{and} \qquad \neg \bigwedge_{i=1}^n \neg w_i \vee F_i \equiv \bigvee_{i=1}^n w_i \wedge \neg F_i$$

Under Gödel semantics, conjunction and disjunction are interpreted as $\min$ and $\max$ respectively, so the two translations become

$$\text{Disj.}: \quad \max_i \min(w_i, F_i), \qquad \text{Conj.}: \quad \min_i \max(1 - w_i, F_i).$$

**Case Study: Selecting Reliable Rules.** Equation 3 proposes the following LoH model, which is also used inside conjunctive neurons of the type in equation 5:

$$\bigwedge_{i=1}^n [r_i, \top] \tag{2}$$

Notice that the two weights in the compilation of a choice operator choosing only among two subformulas must sum to one.[6] Hence, we will write $w_i$ and $1 - w_i$ for the (respective) weights of $r_i$ and $\top$ in $[r_i, \top]$.

With conjunctive compilation, equation 3 yields:

$$\min_i(\min(\max(1 - w_i, r_i), \max(w_i, \top))) = \min_i(\max(1 - w_i, r_i))$$

On the other hand, with disjunctive compilation, it becomes:

$$\min_i(\max(\min(w_i, r_i), \min(1 - w_i, \top))) = \min_i(\max(\min(w_i, r_i), 1 - w_i))$$

---

[6]Indeed, let $z_1$ and $z_2$ be the stored parameters. By design, the weights $w_1$ and $w_2$ are obtained applying the sigmoid function to $z_1 - \frac{z_1 + z_2}{2} = \frac{z_1 - z_2}{2}$ and $z_2 - \frac{z_1 + z_2}{2} = \frac{z_2 - z_1}{2}$ (respectively). The two logit values are opposite to each other, so the weights $w_1 = \sigma(\frac{z_1 - z_2}{2T})$ and $w_2 = \sigma(\frac{z_2 - z_1}{2T})$ must sum to 1.

Thus, equation 3 simplifies more under conjunctive compilation. This simplification has an important effect on the training dynamics. Indeed, when a rule $r_i$ is already (almost) satisfied for a specific example (i.e., $r_i \approx 1$), also the inner $\max(1-w_i, r_i)$ becomes $\approx 1$. Hence, the outer $\min_i$ ignores that index and concentrates on a rule whose truth value is *smaller* (if any). Then, it is the weight associated to that less-satisfied rule that will receive gradient signal for updating. Intuitively, when particular data gives no evidence for preferring $r_i$ over $\top$ (because their truth values are the same), the network also receives no signal to change the associated gate $w_i$. By contrast, $\max(\min(w_i, r_i), 1 - w_i)$ evaluates to $\max(w_i, 1 - w_i)$ when $r_i \approx 1$. Consequently, the loss may push $w_i$ upwards or downwards even when the data provide no information for choosing between $r_i$ and $\top$. These unwanted updates can slow convergence and may bias the learned subset of rules.

**Empirical Evaluation.** We conducted some experiments validating the previous findings and extending to different LoH models. The experiments are conducted in the following way. We consider 10 propositional variables and randomly generate some ground-truth clauses and some additional clauses, of width ranging from 2 to 5. For any of the $2^{10}$ possible Boolean interpretations, a label is produced using the ground-truth clauses as rules. Then the dataset is divided into 75% for training and 25% for evaluation. An LoH model is trained to select some rules among the ground-truth + additional rules. For any considered number of ground-truth clauses and additional clauses, the same experiment is repeated 10 times. For each execution, the test-set F1 score is recorded, together with the number of optimization steps to convergence. The criterion we use for deciding convergence is the following: either 100% accuracy is achieved, or there is no change in accuracy for 64 consecutive steps, or a limit of 64 epochs (384 steps) is reached. The values of the hyperparameters were fixed: 128 as batch size, 0.15 as learning rate, 1 as temperature, and $Gumbel(0, 1)$ noise.

Figure 3 reports the plots with the experiments' results for different models. Subfigure 3a refers to simple rule selection, with LoH models as in equation 3. Subfigure 3b does the same, but in the dual set-up: both ground-truth and additional clauses $c_i$ are *conjunctive* clauses, and the LoH model $\bigvee_{i=1}^{n}[c_i, \bot]$ learns a disjunction of them. Finally, Subfigure 3c considers again rules made of (disjunctive) clauses. However, each of the $m$ ground-truth clauses is placed on a different set $\{r_{i,1}, r_{i,2}, \ldots, r_{i,k+1}\}$, together with $k$ additional clauses. Then, we use the model in equation 4, selecting one rule per set. These plots corroborate our suggestion to use conjunctive compilation inside conjunctions (Subfigures 3a and 3c), and disjunctive compilation inside disjunctions (Subfigure 3b). Indeed, such compilation choices achieved better results both in terms of final accuracy and of convergence speed.

## D  COUNTEREXAMPLE FOR PRODUCT FUZZY LOGIC

Let us consider the disjunctive compilation of $[a, b]$:

$$(w_a \wedge a) \vee (w_b \wedge b)$$

and interpret it with the following fuzzy values: $\mathcal{I}(w_a) = 0.6$, $\mathcal{I}(a) = 0.7$, $\mathcal{I}(w_b) = 0.4$ and $\mathcal{I}(b) = 0$. Using product fuzzy logic,

$$\mathcal{I}((w_a \wedge a) \vee (w_b \wedge b)) = 0.6 * 0.7 = 0.42 < 0.5$$

whose discretized Boolean value $\rho_{0.5}(0.42) = 0$ (False). On the other hand, if $\mathcal{B}$ is the Boolean discretization of $\mathcal{I}$, so that $\mathcal{B}(w_a) = 1$, $\mathcal{B}(a) = 1$, $\mathcal{B}(w_b) = 0$ and $\mathcal{B}(b) = 0$, then

$$\mathcal{B}((w_a \wedge a) \vee (w_b \wedge b)) = 1$$

This means that the discretized explanation disagrees with the behavior of the network. Analogous counterexamples can be built for Łukasiewicz logic. Instead, with Gödel logic, the Boolean interpretation is always in accordance with the fuzzy truth values (Theorem 1):

$$\mathcal{I}((w_a \wedge a) \vee (w_b \wedge b)) = \max(\min(0.6, 0.7), \min(0.4, 0)) = 0.6 > 0.5$$

## E  DETAILS ON THE WILDFIRE RISK ASSESSMENT TASK

**Variables and ground–truth.** The task is defined over nine propositional variables. Two of them are *visual* concepts, to be formed by a CNN from the images: $DenseForest$ and

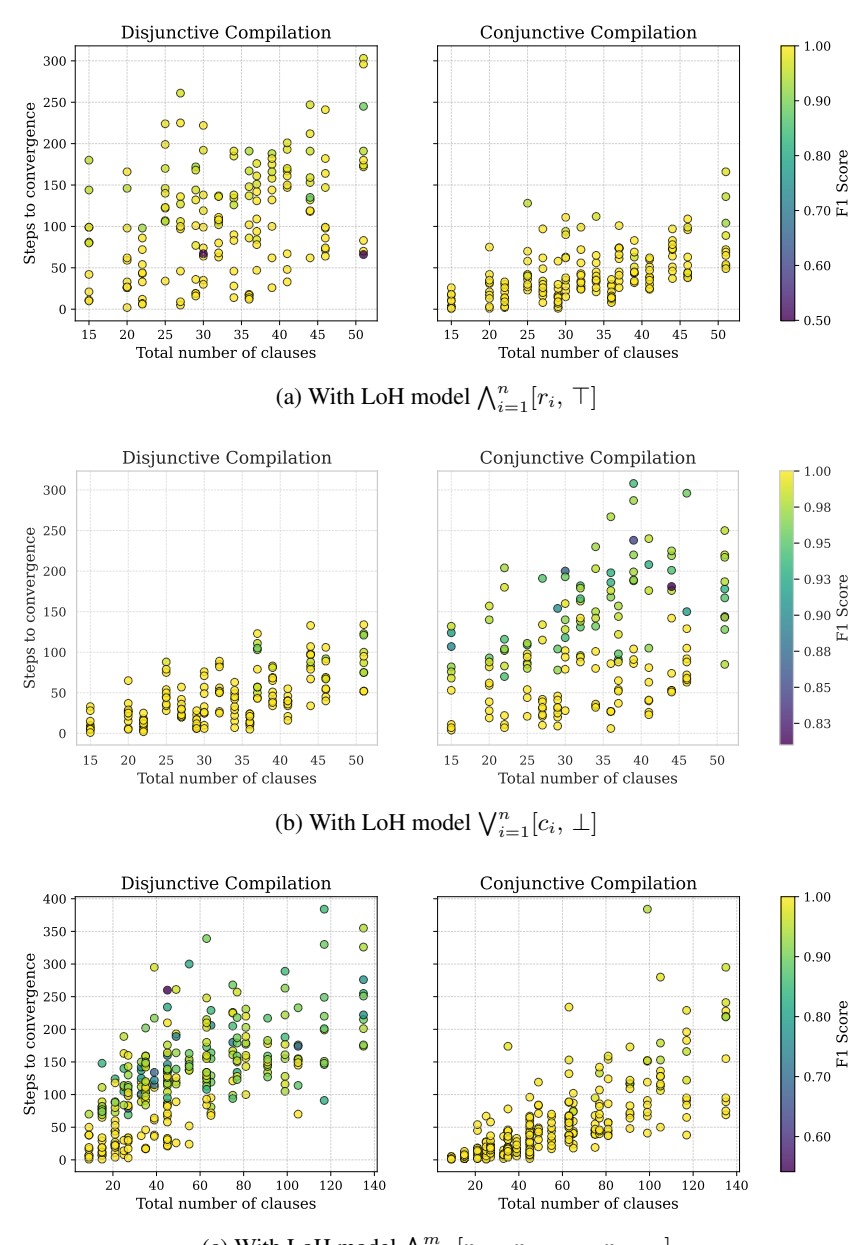

(a) With LoH model $\bigwedge_{i=1}^{n}[r_i, \top]$

(b) With LoH model $\bigvee_{i=1}^{n}[c_i, \bot]$

(c) With LoH model $\bigwedge_{i=1}^{m}[r_{i,1}, r_{i,2}, \ldots, r_{i,k+1}]$

Figure 3: Comparison of disjunctive and conjunctive compilations.

$DryVegetation$. The remaining seven are *non–visual* features provided in tabular form: $LowHumidity$, $HighTemperature$, $RainedRecently$, $StrongWind$, $LightningsFrequent$, $LowHumanActivity$ and $PowerLinesNearby$. The ground–truth label $WildfireRisk$ is obtained by combining three interpretable intermediate rules: see equations 2.

**Dataset generation.** The dataset contains 2048 synthetic RGB images of size $256 \times 256$. For each image, we first sample Boolean assignments to the visual concepts ($DenseForest$ and $DryVegetation$), deciding whether the scene should contain dense forest and whether the vegetation should be dry. A simple drawing routine then renders forest–like patches, and the color scheme distinguish greener from dry vegetation. The image generation process may also produce additional elements, such as houses or rivers, that are random artifact not correlated with the wildfire risk. Examples of the generated images are available in figure 4.

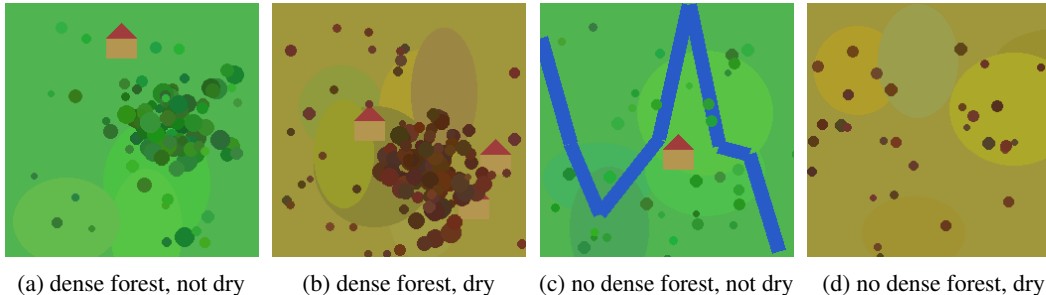

(a) dense forest, not dry    (b) dense forest, dry    (c) no dense forest, not dry    (d) no dense forest, dry

Figure 4: Examples of synthetic wildfire scenes for the four possible combinations of $DenseForest$ and $DryVegetation$.

Independently, we sample Boolean values for the seven non–visual features (e.g. humidity, wind, human activity), as iid $Bernoulli(0.5)$ random variables. The ground–truth wildfire–risk label for each sample is obtained by evaluating the rule above on the full 9–dimensional Boolean vector. The images, the non–visual features and the risk labels are stored in a PyTorch dataset and split into training and test sets (75%/25%).

**Candidate rules.**    Section 6 compares several ways of exploiting symbolic knowledge. In all of them, the starting point is a small pool of candidate rules:

- **Fuel Candidates:**
    - Ground Truth: $DenseForest \lor (DryVegetation \land StrongWind)$
    - Distractors:
        * $DenseForest$
        * $DenseForest \land DryVegetation$
        * $DryVegetation \land (StrongWind \lor LowHumidity)$
- **Dryness Candidates:**
    - Ground Truth: $LowHumidity \lor (HighTemperature \land \neg RainedRecently)$
    - Distractors:
        * $LowHumidity \land HighTemperature \land \neg RainedRecently$
        * $HighTemperature \lor (LowHumidity \land \neg RainedRecently)$
        * $\neg RainedRecently \lor (LowHumidity \land StrongWind)$
        * $\neg RainedRecently \lor (LowHumidity \land HighTemperature)$
        * $DryVegetation \land \neg RainedRecently$
- **Trigger Candidates:**
    - Ground Truth: $LightningsFrequent \lor \neg LowHumanActivity \lor PowerLinesNearby$
    - Distractors:
        * $LightningsFrequent \land LowHumanActivity$
        * $PowerLinesNearby \land StrongWind$
        * $RainedRecently \land LightningsFrequent$
        * $\neg LowHumanActivity$

Here we separate the candidate rules into Fuel, Dryness, and Triggers only for the "Selecting One Rule per Set" knowledge regime. In the "Selecting Reliable Rules" and "Partial Knowledge Base" regimes, all formulas share a single undivided candidate set with no such subdivision.

**Model Architectures and training.**    The neurosymbolic models integrate a perception and a reasoning module, which are trained end-to-end:

- **Perception Module (CNN):** A three-layer Convolutional Neural Network processes $3 \times 256 \times 256$ RGB images to extract the visual predicates $DenseForest$ and $DryVegetation$.

Each block consists of a convolution (channels: $3 \rightarrow 16 \rightarrow 16 \rightarrow 16$), ReLU activation, MaxPool ($2 \times 2$), and Dropout ($p = 0.15$). A final classification head outputs fuzzy values for the two visual concepts, through a *sigmoid* activation.

- **Reasoning Module (LoH):** This layer accepts the visual predictions alongside the non-visual boolean features. It comes in the four different knowledge regimes discussed in Section 6.

Training uses standard binary cross–entropy on the wildfire label, with Adam optimizers for both the logical and perceptual parts. The CNN learning rate is $8 \cdot 10^{-4}$ and the LoH learning rate is $8 \cdot 10^{-2}$. For each knowledge regime, the training is performed 20 times with different random seeds.

## F  COMPARISON OF DIFFERENT TEMPLATES

Let us consider the following ground-truth CNF formula $\phi$ made of 5 definite clauses of width 3:

$$(\neg v_3 \vee \neg v_8 \vee v_7) \wedge (\neg v_{10} \vee \neg v_3 \vee v_4) \wedge (\neg v_1 \vee \neg v_9 \vee v_{10}) \wedge (\neg v_2 \vee \neg v_6 \vee v_8) \wedge (\neg v_4 \vee \neg v_3 \vee v_5)$$

For any of the $2^{10}$ possible Boolean interpretations of the propositional variables $v_1, \ldots, v_{10}$, a ground-truth label is produced using $\phi$. This dataset is divided into $75\%$ for training and $25\%$ for evaluation, and each LoH model is trained to construct 5 clauses. The values of the hyperparameters are fixed: $128$ as batch size, $0.15$ as learning rate, $1$ as temperature, and $Gumbel(0,1)$ noise. Figure 5 compares the average learning curves (over 20 runs) of LoH models adhering to different templates. In particular,

- "5 clauses" uses the conjunction of five copies of $\bigvee_{i=1}^{10}[\neg v_i, v_i, \perp]$
- "5 definite clauses" uses the conjunction of five copies of formula equation 8 with $n = 10$, i.e., $\bigvee_{i=1}^{10}[\neg v_i, \perp] \vee [v_1, \ldots, v_{10}]$
- "5 definite clauses of width 3" uses the conjunction of five copies of the following LoH formula: $[\neg v_1, \ldots, \neg v_{10}] \vee [\neg v_1, \ldots, \neg v_{10}] \vee [v_1, \ldots, v_{10}]$
- "5 definite clauses with heads given" uses $(\bigvee_{i=1}^{10}[\neg v_i, \perp] \vee v_7) \wedge (\bigvee_{i=1}^{10}[\neg v_i, \perp] \vee v_4) \wedge (\bigvee_{i=1}^{10}[\neg v_i, \perp] \vee v_{10}) \wedge (\bigvee_{i=1}^{10}[\neg v_i, \perp] \vee v_8) \wedge (\bigvee_{i=1}^{10}[\neg v_i, \perp] \vee v_5)$
- "5 definite clauses with first given" uses the conjunction of $(\neg v_3 \vee \neg v_8 \vee v_7)$ with four copies of $\bigvee_{i=1}^{10}[\neg v_i, \perp] \vee [v_1, \ldots, v_{10}]$

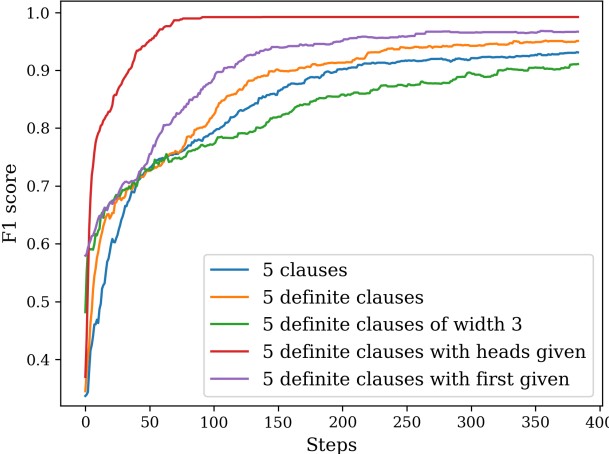

Figure 5: Average training curves — over 20 runs — of LoH formulas following different templates, for learning the same ground-truth CNF formula made of 5 definite clauses of width 3.

We can notice that adding explicit knowledge—fixing either a clause or the clause heads—yield better learning curves than the purely syntactic alternatives. Regarding such three purely syntactic templates, we can notice that the model learning definite clauses slightly outperformed the most general one, thanks to a reduced search space. However, despite having an even smaller hypothesis space, the

model learning definite clauses of width 3 is the one performing worse. This may be explained by the fact that the other models can update the weights of the negative literals independently, whereas updates in $[\neg v_1, \ldots, \neg v_{10}]$ always involve more variables at the same time. Additionally, choosing $\neg v_i$ in the first choice operator and $\neg v_j$ in the second (with $i \neq j$) is in a completely different region of the search space from the equivalent choice of $\neg v_j$ in the first and $\neg v_i$ in the second. Anyway, one may still employ this model because it guarantees formulas of a prescribed template, irrespective of raw predictive performance. Moreover, among the 20 runs, each of the LoH models found the 100%-correct formula multiple times. This suggest that trying different runs and picking the best can be a useful strategy.

## G   FURTHER EXPERIMENTS WITH ARTIFICIAL DATA

Figure 6 reprises the experiments of Appendix C. However, it considers only the best compilations (disjunctive for 6b, and conjunctive for 6a and 6c), while highlighting the influence of different experimental factors. In particular, the middle column shows how both F1 score and convergence speed have a strong negative correlation with the number of ground-truth clauses. Clearly, the more clauses in the ground truth, the more need to be selected for a perfect score, and the more difficult the learning. However, since the ground-truth clauses are independently generated, their number is also strongly correlated with data imbalance. As shown in the first column, learning a conjunction of clauses suffers most when there are too few positive samples (i.e., samples in which the ground-truth formula is satisfied). Conversely, a shortage of negative samples drives poorer performance when learning a disjunction. In contrast, the number of additional, "misleading" clauses is not as impactful as the ground-truth, as evidenced by the third column. This is true at least when the number of additional clauses is comparable to that of the ground truth.[7]

## H   DATASETS PROPERTIES

For each dataset, Table 3 summarizes the number of features before and after binarization, together with references and size.

Table 3: Datasets properties.

| Dataset | # instances | # classes | # original features | # binary features |
|---|---|---|---|---|
| adult (Becker & Kohavi, 1996) | 32561 | 2 | 14 | 155 |
| bank-marketing (Moro et al., 2014) | 45211 | 2 | 16 | 88 |
| banknote (Lohweg, 2012) | 1372 | 2 | 4 | 17 |
| blogger (blo, 2012) | 100 | 2 | 5 | 15 |
| chess (Bain & Hoff, 1994) | 28056 | 18 | 6 | 40 |
| connect-4 (Tromp, 1995) | 67557 | 3 | 42 | 126 |
| letRecog (Slate, 1991) | 20000 | 26 | 16 | 155 |
| magic04 (Bock, 2004) | 19020 | 2 | 10 | 79 |
| mushroom (Schlimmer, 1981) | 8124 | 2 | 22 | 117 |
| nursery (Rajkovic, 1989) | 12960 | 5 | 8 | 27 |
| tic-tac-toe (Aha, 1991) | 958 | 2 | 9 | 27 |
| wine (Cortez et al., 2009) | 178 | 3 | 13 | 37 |

## I   HYPERPARAMETERS

The following outlines the hyperparameter search spaces for the models presented in Section 7. The hyperparameter choices for each dataset are available in the code repository.

- Decision Tree: min_samples_split (2–50), max_depth (2–50), min_samples_leaf (1–50).
- Random Forest: n_estimators (50–500), min_samples_split (2–50), max_depth (2–50), min_samples_leaf (1–50).

---

[7]Regarding Subfigure 6c, notice that the *total* number of added clauses is a *multiple* of the number of additional clauses *per set*.

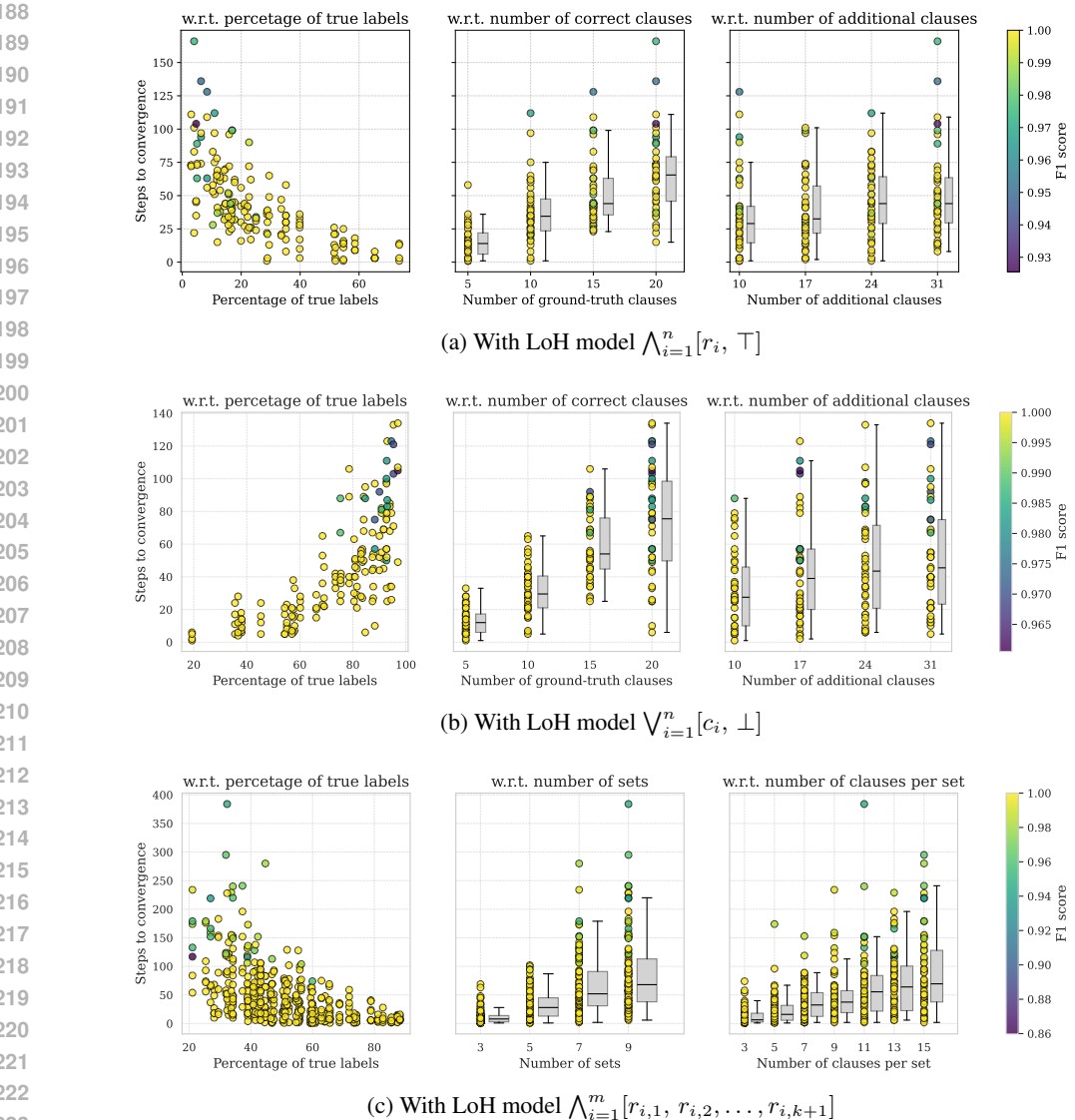

(a) With LoH model $\bigwedge_{i=1}^{n}[r_i,\ \top]$

(b) With LoH model $\bigvee_{i=1}^{n}[c_i,\ \bot]$

(c) With LoH model $\bigwedge_{i=1}^{m}[r_{i,1},\ r_{i,2},\dots,r_{i,k+1}]$

Figure 6: Clauses selection performance w.r.t. percentage of true labels, number of ground-truth clauses and number of additional clauses.

- XGBoost: max_depth (5–20), n_estimators (10–500), learning_rate ($10^{-3}$–$2\cdot10^{-1}$).

- Neural Network: number of hidden layers (1–3), number of units per layer (4–128), learning rate ($10^{-4}$–$10^{-1}$). Batch size fixed at 256, and ReLU activations.

- DLN: number of hidden layers (1–10), number of units per layer (16–512), grad_factor (1.–2.), learning rate ($10^{-3}$–$10^{-1}$), $\tau$ (1–100). Batch size fixed at 128.

- MLLP/CRS: number of hidden layers (1,3), number of units per layer (16–256), weight decay ($10^{-8}$–$10^{-2}$), learning rate ($10^{-4}$–$10^{-1}$), random binarization rate (0–0.99). Batch size (128) and learning rate scheduler were set to the default value.

- Ours: learning rate (0.01–0.2), Gumbel noise scale $\beta$ (0.4–1.2), temperature (0.4–1.2), and temperature rescaling factor applied every 10 epochs (0.9925–1). Like NN and MLLP/CRS, we also optimized the architecture, allowing the TPE algorithm to select the layer type (any–clause vs fixed–size–clauses), the number of hidden layers (1–2), layer sizes (16–256), and whether the output layer is conjunctive or disjunctive (with the other layers alternating). In case of fixed–size–clauses layer type, also the hyperparameter $k$ (2–8) was tuned for each layer.

## J  Decision Rules of Visual Tic-Tac-Toe

Both CRS and LoH allow for the automatic extraction of logical formulas. In order to interpret such decision rules, we need to assign proper names to the input propositions. In particular, for visual tic-tac-toe, there are three input propositions for every image in the tic-tac-toe grid, each corresponding to an output unit of the feature-extractor CNN. The assignment of labels (such as $X$, $O$ or $B$) to such units is done by thresholding their average activations with respect to each image class (0, 1, and 2), in the following way:

- if the average activation is never $> 0.5$, the label for the unit is $\bot$, which can later be simplified from the formulas;
- if the average activation is $> 0.5$ for one class and $< 0.5$ for the other two, the label for the unit is the one corresponding to the activating class;
- if the average activation is $> 0.5$ for two classes and $< 0.5$ for the other, the label for the unit is the logical negation of the non-activating class;
- if the average activation is always $> 0.5$, the label for the unit is $\top$, which can later be simplified from the formulas.

As an example, if an output unit of the CNN exhibits an average activation below $0.5$ for images of class 1 but above $0.5$ for images of the other two classes, we assign it the label $\neg O$ (recalling that digit 1 was associated with $O$). In this way, we can assign names to the input propositions of the logical models, by combining the labels of the CNN output units with the positions in the $3 \times 3$ tic-tac-toe grid.

Table 4 reports the *best* decision rules learned by CRS and our base model on the Visual Tic-Tac-Toe task. The formulas were simplified removing the appearances of $\top$ and $\bot$, and also removing redundant clauses. We do not provide the formulas learned by DLN because the implementation of DLN we used did not have a function to write them in a human-readable way. Moreover, to boost performance, DLN actually learns an ensemble with majority voting, not a single formula. The DLN run with highest Symbolic eval achieved .885 F1-score.

## K  Runtimes

Table 5 reports the approximated runtime of a single training plus evaluation run, for each benchmark discussed in Section 7. The values are relative to the selected hyperparameters, and the table also reports the corresponding models' parameter count. All experiments were conducted on a cluster node equipped with an Nvidia RTX A5000 with 60GB RAM.

Table 4: Best decision rules learned on the Visual Tic-Tac-Toe task. The labels $X_i$ and $\neg O_j$ were assigned to each proposition in the way explained in this appendix.

| Model | Symb. eval | | Formula |
|---|---|---|---|
| CRS | DNF | .991 | $(\neg O_1 \wedge \neg O_5 \wedge \neg O_9) \vee (\neg O_3 \wedge \neg O_5 \wedge \neg O_7)$ 
 $\vee(\neg O_2 \wedge \neg O_4 \wedge \neg O_7 \wedge \neg O_9) \vee (\neg O_1 \wedge \neg O_2 \wedge \neg O_3 \wedge \neg O_4 \wedge \neg O_7)$ 
 $\vee(\neg O_1 \wedge \neg O_2 \wedge \neg O_3 \wedge \neg O_4 \wedge \neg O_8) \vee (\neg O_1 \wedge \neg O_2 \wedge \neg O_3 \wedge \neg O_4 \wedge \neg O_9)$ 
 $\vee(\neg O_1 \wedge \neg O_2 \wedge \neg O_3 \wedge \neg O_5 \wedge \neg O_8) \vee (\neg O_1 \wedge \neg O_2 \wedge \neg O_3 \wedge \neg O_6 \wedge \neg O_7)$ 
 $\vee(\neg O_1 \wedge \neg O_2 \wedge \neg O_3 \wedge \neg O_6 \wedge \neg O_8) \vee (\neg O_1 \wedge \neg O_2 \wedge \neg O_4 \wedge \neg O_6 \wedge \neg O_7)$ 
 $\vee(\neg O_1 \wedge \neg O_3 \wedge \neg O_4 \wedge \neg O_7 \wedge \neg O_8) \vee (\neg O_1 \wedge \neg O_3 \wedge \neg O_6 \wedge \neg O_8 \wedge \neg O_9)$ 
 $\vee(\neg O_1 \wedge \neg O_4 \wedge \neg O_5 \wedge \neg O_6 \wedge \neg O_7) \vee (\neg O_1 \wedge \neg O_4 \wedge \neg O_5 \wedge \neg O_6 \wedge \neg O_8)$ 
 $\vee(\neg O_1 \wedge \neg O_4 \wedge \neg O_7 \wedge \neg O_8 \wedge \neg O_9) \vee (\neg O_1 \wedge \neg O_6 \wedge \neg O_7 \wedge \neg O_8 \wedge \neg O_9)$ 
 $\vee(\neg O_2 \wedge \neg O_3 \wedge \neg O_4 \wedge \neg O_5 \wedge \neg O_8) \vee (\neg O_2 \wedge \neg O_3 \wedge \neg O_4 \wedge \neg O_6 \wedge \neg O_9)$ 
 $\vee(\neg O_2 \wedge \neg O_3 \wedge \neg O_6 \wedge \neg O_7 \wedge \neg O_9) \vee (\neg O_2 \wedge \neg O_4 \wedge \neg O_5 \wedge \neg O_6 \wedge \neg O_8)$ 
 $\vee(\neg O_2 \wedge \neg O_4 \wedge \neg O_5 \wedge \neg O_6 \wedge \neg O_9) \vee (\neg O_2 \wedge \neg O_4 \wedge \neg O_5 \wedge \neg O_8 \wedge \neg O_9)$ 
 $\vee(\neg O_2 \wedge \neg O_5 \wedge \neg O_6 \wedge \neg O_7 \wedge \neg O_8) \vee (\neg O_2 \wedge \neg O_5 \wedge \neg O_7 \wedge \neg O_8 \wedge \neg O_9)$ 
 $\vee(\neg O_3 \wedge \neg O_4 \wedge \neg O_5 \wedge \neg O_6 \wedge \neg O_8) \vee (\neg O_3 \wedge \neg O_4 \wedge \neg O_5 \wedge \neg O_6 \wedge \neg O_9)$ 
 $\vee(\neg O_3 \wedge \neg O_4 \wedge \neg O_6 \wedge \neg O_8 \wedge \neg O_9) \vee (\neg O_3 \wedge \neg O_4 \wedge \neg O_7 \wedge \neg O_8 \wedge \neg O_9)$ 
 $\vee(\neg O_3 \wedge \neg O_6 \wedge \neg O_7 \wedge \neg O_8 \wedge \neg O_9)$ |
| CRS | CNF | .984 | $(\neg O_1 \vee \neg O_2 \vee \neg O_3) \wedge (\neg O_1 \vee \neg O_4 \vee \neg O_7)$ 
 $\wedge(\neg O_1 \vee \neg O_5 \vee \neg O_9) \wedge (\neg O_2 \vee \neg O_5 \vee \neg O_8)$ 
 $\wedge(\neg O_3 \vee \neg O_5 \vee \neg O_7) \wedge (\neg O_3 \vee \neg O_6 \vee \neg O_9)$ 
 $\wedge(\neg O_4 \vee \neg O_5 \vee \neg O_6) \wedge (\neg O_7 \vee \neg O_8 \vee \neg O_9)$ 
 $\wedge(\neg O_2 \vee \neg O_3 \vee \neg O_4 \vee \neg O_9) \wedge (\neg O_2 \vee \neg O_5 \vee \neg O_7 \vee \neg O_9)$ |
| Ours | DNF | 1.00 | $(X_1 \wedge X_2 \wedge X_3) \vee (X_4 \wedge X_5 \wedge X_6)$ 
 $\vee(X_7 \wedge X_8 \wedge X_9) \vee (X_1 \wedge X_4 \wedge X_7)$ 
 $\vee(X_2 \wedge X_5 \wedge X_8) \vee (X_3 \wedge X_6 \wedge X_9)$ 
 $\vee(X_1 \wedge X_5 \wedge X_9) \vee (X_3 \wedge X_5 \wedge X_7)$ |
| Ours | CNF | .999 | $(X_1 \vee X_5 \vee X_9) \wedge (X_3 \vee X_5 \vee X_7)$ 
 $\wedge(X_1 \vee X_2 \vee X_6 \vee X_7) \wedge (X_1 \vee X_3 \vee X_4 \vee X_8)$ 
 $\wedge(X_1 \vee X_3 \vee X_5 \vee X_8) \wedge (X_1 \vee X_3 \vee X_6 \vee X_8)$ 
 $\wedge(X_1 \vee X_5 \vee X_6 \vee X_7) \wedge (X_1 \vee X_6 \vee X_7 \vee X_8)$ 
 $\wedge(X_2 \vee X_3 \vee X_4 \vee X_9) \wedge (X_2 \vee X_4 \vee X_5 \vee X_9)$ 
 $\wedge(X_2 \vee X_4 \vee X_7 \vee X_9) \wedge (X_2 \vee X_5 \vee X_6 \vee X_7)$ 
 $\wedge(X_2 \vee X_5 \vee X_7 \vee X_9) \wedge (X_2 \vee X_6 \vee X_7 \vee X_9)$ 
 $\wedge(X_3 \vee X_4 \vee X_5 \vee X_8) \wedge (X_3 \vee X_4 \vee X_5 \vee X_9)$ 
 $\wedge(X_3 \vee X_4 \vee X_8 \vee X_9) \wedge (X_1 \vee X_2 \vee X_3 \vee X_4 \vee X_7)$ 
 $\wedge(X_1 \vee X_2 \vee X_3 \vee X_6 \vee X_9) \wedge (X_1 \vee X_2 \vee X_5 \vee X_6 \vee X_8)$ 
 $\wedge(X_1 \vee X_4 \vee X_5 \vee X_6 \vee X_8) \wedge (X_1 \vee X_4 \vee X_7 \vee X_8 \vee X_9)$ 
 $\wedge(X_2 \vee X_4 \vee X_5 \vee X_6 \vee X_8) \wedge (X_3 \vee X_6 \vee X_7 \vee X_8 \vee X_9)$ |

Table 5: Average runtimes relative to a single training + evaluation run.

| Dataset | DLN | | MLLP | | Ours | |
|---|---|---|---|---|---|---|
| | Time (s) | # Params | Time (s) | # Params | Time (s) | # Params |
| adult | 520 | 57680 | 499 | 74260 | 287 | 17584 |
| bank-marketing | 615 | 38576 | 687 | 70556 | 452 | 81612 |
| banknote | 17 | 23728 | 14 | 4807 | 12 | 7790 |
| blogger | 4 | 36592 | 4 | 3485 | 6 | 6195 |
| chess | 956 | 53568 | 970 | 117774 | 546 | 29464 |
| connect-4 | 1214 | 51344 | 650 | 25542 | 873 | 55212 |
| letRecog | 414 | 41280 | 511 | 126302 | 528 | 86518 |
| magic04 | 467 | 50960 | 271 | 19440 | 364 | 31104 |
| mushroom | 110 | 31024 | 78 | 16422 | 67 | 82900 |
| nursery | 284 | 51840 | 302 | 48331 | 311 | 68186 |
| tic-tac-toe | 25 | 18288 | 38 | 4002 | 32 | 11310 |
| wine | 2 | 6960 | 5 | 6400 | 4 | 2640 |

