# OpenReview forum: "Logic of Hypotheses: From Zero to Full Knowledge in Neurosymbolic Integration"
_ICLR.cc/2026/Conference — Submitted to ICLR 2026_

### Official Review · Reviewer_Acyb · 2025-10-31

**Soundness:** 2
**Presentation:** 3
**Contribution:** 1
**Rating:** 2
**Confidence:** 3

**Summary:**

This paper introduces Logic of Hypotheses (LoH), a neurosymbolic framework that extends propositional logic with a choice operator allowing the model to select among alternative subformulas in a differentiable way. Using Gödel fuzzy semantics and a reparameterization mechanism inspired by the Gödel trick, the authors claim that LoH can unify both knowledge injection and knowledge extraction within a single formalism.

**Strengths:**

The manuscript is well written.

The grounding in Gödel fuzzy logic and the preservation of logical consistency is elegant

**Weaknesses:**

Conceptual novelty and positioning is weak

Unclear novelty of choice operator

Unclear scope of contribution

Unsupported interpretability claims

I will describe these points below.

**Questions:**

**Conceptual novelty and positioning**
The claimed unification of knowledge injection and knowledge extraction is not clearly distinct from what existing (neuro)symbolic systems already achieve. Every ILP-style method naturally supports both the inclusion of background knowledge and the induction of new rules, as once worked the semantics of the rules-to-learn adding rules is automatic. As a result, the conceptual advance over prior work remains ambiguous.

**Semantic grounding of the “choice” operator**
The proposed operator resembles constructs that are well established in Statistical Relational AI and Probabilistic Logic Programming, such as probabilistic facts, annotated disjunctions and random-variable choices. The paper does not discuss these connections nor explain whether LoH provides new semantics beyond these long-known interpretations of discrete uncertainty. Given that “choosing among alternatives” is central to many probabilistic logic models, this omission weakens the originality and theoretical depth of the proposal.

**Unclear scope of contribution (syntax vs. semantics)**
Connecting to the point before, it is difficult to determine whether LoH represents a genuinely new semantic model or merely a syntactic/implementation-level reformulation of existing differentiable logic frameworks. The paper does not demonstrate that LoH achieves behaviors that could not already be implemented within DeepProbLog, LTN or other differentiable logical systems by suitable parameterization. The lack of such differentiation undermines the claimed unification.

**Interpretability claims**
In the experiments, the results primarily compare predictive accuracy and do not substantiate the key interpretability or symbolic extraction claims. When used on a parallel and deep extraction of rules, such architecture does not seem to be any more interpretable than existing neural models, despite maybe the discrete nature, which is not a prerogative of the proposed method. Therefore, interpretability claims are nos]t supported by sufficient evidence.

---

> ### Author Response · Authors · 2025-11-25
>
> **Conceptual novelty and positioning (Question).**
> We agree that ILP systems, including differentiable ILP, can already combine injected background knowledge with rule induction.
> LoH, however, is designed in the style of the differentiable logic-based NeSy architectures that are most commonly used in our target applications: systems such as LTN and DeepProbLog on the "knowledge-injection" side, and propositional architectures such as MLLP, Net-DNF and DLN on the "rule-mining" side.
> LoH provides a single differentiable formalism that spans these regimes, by offering a novel and unified way to design fuzzy logic neural networks whose structure can be flexibly tailored (incorporating given rules and partially filled, or void, arbitrary templates), all with faithful Boolean rule extraction.
> This setting differs from ILP not only in being propositional but in design paradigm: instead of meta-templates, clause enumeration, mode declarations, auxiliary predicates and the background vs hypothesis split, LoH represents everything as a single choice-augmented formula whose learnable choices directly capture the parts to be induced.
> We may revise the manuscript to explicitly restrict the scope of the unification claim to this non-relational NeSy landscape, and to clarify ILP capabilities.
>
> Regarding ILP, differentiable frameworks like $\partial$ILP first enumerate all clauses allowed by a fixed template and then assign a continuous weight to each one. This poses limitations in terms of scalability and require the use of many additional predicates to achieve flexibility. On the other hand, symbolic ILP engines such as Metagol provide more flexible second‑order templates but are not differentiable, so they cannot be trained jointly with perception modules. Some newer ILP methods like Neural Rule Learner (NeurRL) do not even support user-provided background rules in their implementation. Together, these properties may let LoH fill a gap also between current ILP systems: it is differentiable as $\partial$ILP, but it offers great flexibility in terms of how the formulas to be learned can be written. All while offering faithful rule extraction (thanks to Gödel semantics) and without sacrificing scalability.
>
> **Semantic grounding of the “choice” operator (Question).**
> We thank the reviewer for the insightful question of better relating our framework and probabilistic logic programming / statistical relational learning.
> In PLP and SRL, probabilistic facts and annotated disjunctions define random variables over ground atoms or rule heads, and their semantics is a probability measure over interpretations (possible worlds).
> In contrast, a compiled LoH formula act as one formula in the hypothesis space, making binary deterministic yes/no predictions: the one obtained substituting each choice operator with the subformula with weight $>0.5$. LoH training is searching for this single global formula that best explains the data, with the choice operators acting as placeholders for the subformulas during learning, and the fuzzy values as a sort of confidence margins. This process can be interpreted as a (differentiable) combinatorial local search algorithm [1].
>
> Thus, probabilistic facts / annotated disjunctions introduce stochasticity over _worlds_ given a fixed theory. On the other hand, the choice operator introduces structural variability over _theories_, and is used to select a single preferred theory from a finite hypothesis space, with a combinatorial search.
>
> [1] Daniele, Alessandro, and Emile van Krieken. "Noise to the Rescue: Escaping Local Minima in Neurosymbolic Local Search." arXiv preprint (2025).

---

> ### Author Response · Authors · 2025-11-25
>
> **Unclear scope of contribution (syntax vs. semantics) (Question).**
> Connected to the point before about the semantic grounding of the choice operator, notice that pulling out one crisp formula from a probabilistic model may be problematic unless every probability collapses to $0$ or $1$ on the trained model. In fact, picking the most likely branch for each local choice results in a heuristic approximation without guarantees. LoH avoids this issue: by construction it always ends with one deterministic formula whose predictions exactly match the trained network.
> This makes LoH suited for tasks like rule learning or deterministic decision modeling, whereas probabilistic logic programs excel at generative or probabilistic inference tasks (answering queries with probabilities given uncertain facts).
>
> Indeed, to our knowledge, systems like LTN, DeepProbLog, DeepStochLog, NeurASP, PRISM, etc., were never applied in rule learning settings like the ones in our experiments. Reprogramming their parameters for learning the symbolic rules instead of the possible worlds could be an interesting research direction which, to our knowledge, is not currently addressed. For example, the conclusion of the paper introducing DeepStochLog (which uses annotated disjunctions) says "There are several limitations of DeepStochLog that we want to explore in further research. First, DeepStochLog does not yet learn the structure of the rules..." [2].
>
> In both LTN and DeepProbLog, the logical theory is fixed by the user. The parameters interpreted as probabilities (or fuzzy values) sit only on the outputs of neural predicates/annotated facts, never on alternative rule bodies. Consequently LTN and DeepProbLog can combine perception with probabilistic reasoning, but cannot learn logical rules as the choice operator does. When DeepProbLog is asked to perform "program induction" (as in experiment T3, T4 and T5 in [3]), the logical rules are provided and what is actually induced are internal subsymbolic neural modules. Similar subsymbolic modules could be inserted also inside LoH models, but the point of the choice operator is to learn explicit symbolic rules.
>
> **Interpretability Claims (Question).**
> Following Lipton's taxonomy of transparency [4], even when a symbolic model is too complicated to be well-understood globally (lacking _simulatability_), its discrete nature helps _decomposability_ (i.e., intelligibility of the internal individual components).  This make symbolic models inherently more interpretable than similarly large subsymbolic networks whose individual weights and activations do not have direct meanings.
> In our experiment, the neural network learned to perfectly predict the symbolic tic-tac-toe dataset, but no human could ever understand anything about tic-tac-toe winning conditions just by looking at its weight matrices. On the other hand, a DNF formula -though possibly long- is simply a list of predicted winning conditions and can be directly interpreted.
>
> What usually hampers more the interpretability of Boolean formulas is excessive depth and the use of negations, and we took care to limit both. Regarding length, any post-hoc pruning method (starting from simple redundant rules detection) may be used for its reduction. Nonetheless, we understand that all these aspects deserve more quantitative support, and thanks the reviewer for pointing it out. We are working to report the statistics of the sizes and structures of the formulas, for our models and the baselines.
>
> [2] Winters, Thomas, et al. "Deepstochlog: Neural stochastic logic programming." Proceedings of the AAAI Conference on Artificial Intelligence. Vol. 36. No. 9. 2022.
>
> [3] Manhaeve, Robin, et al. "Deepproblog: Neural probabilistic logic programming." Advances in neural information processing systems 31 (2018).
>
> [4] Lipton, Zachary C. "The mythos of model interpretability: In machine learning, the concept of interpretability is both important and slippery." Queue 16.3 (2018): 31-57.

---

### Official Review · Reviewer_UUDb · 2025-11-01

**Soundness:** 3
**Presentation:** 3
**Contribution:** 3
**Rating:** 6
**Confidence:** 3

**Summary:**

This paper introduces the Logic of Hypotheses (LoH), a novel language designed to unify the two primary paradigms of neurosymbolic integration: injecting pre-defined rules and inducing rules from data. LoH syntactically extends propositional logic with a choice operator, which compactly represents a hypothesis space, i.e., a set of candidate formulas. The key mechanism involves compiling any LoH formula into a differentiable computational graph using fuzzy logic, enabling the selection of the optimal formula from this space to be learned via standard backpropagation. By uniquely leveraging Gödel fuzzy logic and the Gödel trick, the resulting model can be discretized into a hard Boolean-valued function without loss in performance. The framework demonstrates strong performance on tabular data and a visual Tic-Tac-Toe benchmark.

**Strengths:**

- This paper elegantly bridges the gap between knowledge injection and rule induction paradigms in neurosymbolic AI, offering a single formalism that handles a spectrum of prior knowledge specification.
- The language of Logic of Hypotheses (LoH) hinges on the choice operator, which provides flexibility for encoding hypothesis spaces, syntactic templates (CNF, DNF, Horn clauses), and partial knowledge scenarios.
- Furthermore, LoH formulas can be compiled into a fully differentiable computation graph. This allows the model to be trained with backpropagation and stacked seamlessly on top of neural networks (like a CNN) to enable end-to-end learning from raw data (like images) to symbolic rules.

**Weaknesses:**

- The framework is restricted to propositional logic, limiting applicability to relational/structured domains. After all, many real-world NeSy applications require relational reasoning.
- The scalability of the paper to multi-class classification tasks is weak. The experimental results show that while LoH performs well on binary and few-class tasks, it struggles on datasets with a large number of classes.
- Although the paper spends many efforts describing LoH's flexibility in handling partial knowledge (e.g., revising rules, using templates), the main experiments in Section 7 focus on the zero-knowledge (pure rule induction) setting.
- (minor) Figure 2-4: Consider moving some to the main text; they provide valuable insights.

**Questions:**

- When extending to FOL as a next step, what are the biggest challenges in doing this? How would the choice operator concept be extended to handle variable quantifiers?
- The reparameterization for multi-class (Section 7.1) seems ad-hoc. Have you considered alternative approaches (e.g., one-vs-rest, hierarchical)?

---

> ### Author Response · Authors · 2025-11-25
>
> **Extension to FOL (Question).** We agree that many NeSy applications are relational. However, most NeSy systems (e.g., LTN, DeepProbLog, $\partial$ILP, ...) already work by first grounding over a finite domain and then running a differentiable rule module on the resulting propositional setting.
> The genuinely first-order part is the aggregation over the different groundings/substitutions, connected to the quantifiers.
> A first-order version of LoH would bring these grounding and aggregation steps into the logic itself. Concretely, one could enrich the language with $\forall$ and $\exists$, interpreted (in the usual finite-domain setting) as conjunctions/disjunctions over groundings. Then, choice operators could range over both quantified and quantifier-free formulas. The main limitation would be the combinatorial blow-up of groundings, as in many NeSy and Statistical Relational Learning systems based on FOL.
>
> **Multiclass classification (Question).** We recognize that the multi-class performance was a limitation in the current experimental section. And you correctly noted that the multi-class reparameterization in Section 7.1 appears ad-hoc. The motivation was to have something that, unlike a standard softmax, always outputs one (and only one) value above the $0.5$ threshold. We did not think about alternatives like one-vs-one and one-vs-rest, but it makes sense to try. Thank you for the idea.
>
> **Partial knowledge experiments.** Thank you for highlighting the importance of better demonstrating the partial-knowledge setting. We are extending the paper with additional experiments on a synthetic dataset that explicitly test the different use cases discussed in Section 6, in scenarios with partial and full prior knowledge.

---

### Official Review · Reviewer_Enp6 · 2025-11-01

**Soundness:** 2
**Presentation:** 3
**Contribution:** 2
**Rating:** 6
**Confidence:** 3

**Summary:**

This paper proposes a unified training framework that bridges the two main directions in NeSy: knowledge injection and rule induction. Building on Gödel fuzzy logic, the authors extend propositional logic by introducing a choice operator and assigning learnable weights to subformulas, enabling the model to learn logical structures that range from zero prior knowledge to full prior knowledge. The resulting language is differentiable while preserving discrete Boolean semantics, thus supporting supervised gradient-based learning. Experimental results on small-scale tasks, including tabular datasets and a visual tic-tac-toe setting, demonstrate promising performance and interpretable logical representations.

**Strengths:**

1. The introduction of the choice operator allows the model to flexibly handle scenarios ranging from zero to full knowledge and supports different search tendencies, showing strong generality.
2. Under Gödel semantics, the proposed framework achieves differentiability without losing Boolean precision, with rigorous and self-consistent theoretical derivations.
3. The paper provides both analytical and empirical comparisons between disjunctive and conjunctive compilations, clarifying their respective applicability and showing that the approach is practically feasible and adaptable to data characteristics.

**Weaknesses:**

1. The experimental setups are relatively simple. For instance, the tic-tac-toe task only involves nine propositional variables, and the target logic can be expressed using a few disjunctive or conjunctive clauses. The framework’s computational scalability in larger logical spaces has not been demonstrated.
2. The paper directly adopts Gödel fuzzy logic without a clear justification of its advantages compared to alternative fuzzy logics such as Łukasiewicz or Product logic.
3. Although the framework claims to handle the full spectrum from zero knowledge to full knowledge, all experiments are conducted under the zero-knowledge setting; results for partial- and full-knowledge scenarios are missing.
4. The reparameterization of weights as continuous parameters z_i(logits) lacks a theoretical analysis of feasibility, stability, or convergence during training.

**Questions:**

If the visual tic-tac-toe input is first processed by a CNN to produce propositional variables, what is the motivation for using visual data instead of directly symbolic inputs? It would be helpful for the authors to further clarify this design choice.

---

> ### Author Response · Authors · 2025-11-25
>
> **Scalability.** We acknowledge the reviewer's point that our current experiments are limited for analyzing the full  scalability to very large logical spaces. However, even the visual Tic-Tac-Toe task, despite having a relatively simple underlying logic, proved challenging for comparable state-of-the-art neurosymbolic models.
>
> **Justification of Gödel logic usage.** As you mentioned in the strengths, "under Gödel semantics, the proposed framework achieves differentiability without losing Boolean precision, with rigorous and self-consistent theoretical derivations." Your intuition perfectly summarize one of the motivations of our approach.
> This is precisely the reason we adopted Gödel logic over alternatives like Łukasiewicz or Product logic, as discussed in the paragraph "Robustness to Binarization" (page 5). Moreover, appendix D provides an example of failure of product logic to binarization.
>
> **Experiments with full and partial knowledge.**  Thank you for pointing out the lack of explicit results covering the full spectrum of knowledge integration.
> Appendix E provides a comparison of different templates for recovering a simple ground-truth formula, which directly relates to specifying different levels of prior knowledge. Some further experiments with partial knowledge are available also in appendix F.
> However, we understand your critique, and agree that additional experiments would strengthen our claim. To this end, we are extending the paper with additional experiments on a synthetic dataset to test each use case highlighted in Section 6.
>
> **Reparametrization.** The properties of the reparametrization trick are derived from [1], which frames its optimization as a stochastic local search. Appendix A also reports the proof of its equivalence to the Gumbel-max Trick, of which the well-known Gumbel-softmax Trick is an approximation. We would be really thankful to the reviewer if they could better specify, in practice, the empirical or theoretical analysis they would like us to perform.
>
> [1] Daniele, Alessandro, and Emile van Krieken. "Noise to the Rescue: Escaping Local Minima in Neurosymbolic Local Search." arXiv preprint (2025).
>
> **Visual tic-tac-toe (Question).** Thank you for the question; we will try to rephrase the discussion of the design of the experiment to make it clearer. The motivation of the design choice is to provide an instance of the symbol grounding problem, where the model (CNN + logical network) has to ground the symbolic part over the continuous high-dimensional perceptions. Note that the CNN is not pretrained to recognize the digits, nor are its outputs predefined to represent anything specific. Instead, the CNN is trained together with the rule-learning logical network, in an end-to-end way. The only supervision is whether $X$ is winning or not at tic-tac-toe. Thus, we show that LoH can successfully backpropagate gradients through the discrete symbols to the perception layers, allowing the CNN to learn to produce meaningful outputs solely based on the logical signal on top.

---

### Official Review · Reviewer_wHxv · 2025-11-05

**Soundness:** 2
**Presentation:** 1
**Contribution:** 2
**Rating:** 2
**Confidence:** 3

**Summary:**

The paper proposes (i) a new language that extends propositional logic with a choice operator, (ii) a compilation procedure to transform any LoH formula into a differentiable computational graph and (iii) a unifying viewpoint of NeSy paradigms.

**Strengths:**

The paper presents a very interesting method which appears to be novel

**Weaknesses:**

The notation seems a bit imprecise, especially in the definition of the Logic of Hypothesis language especially from the main text. Below I list some questions/imprecisions that I think need some improvement:

1. Shouldn’t the grammar include $\top$ and $\bot$ to make propositional formulas well-formed? Otherwise the inductive definition has no base case except variables. Is this intended? If so why? and where is this specified? The authors write that the choice operator can take any finite number of formulas. What would [] mean?

2. Connected to the above: the authors at the bottom of page 4 write that "LoH is flexible enough to encode any finite set of propositional logic formulae" how would you encode $\{\top\}$?

3. The authors never formally define the concept of hypothesis space

4. The LoH formulas are sometimes interpreted as a set and sometimes as a formula in R1-R5. I think this makes the definitions imprecise

5. I am really struggling to see why you introduce the choice operator if then sometimes you treat it as conjunction and sometimes as disjunction. Wouldn't it be cleaner to treat the problem in propositional logic and simply compile it from there? What is the real advantage of having this operator?

6. I struggle to follow example 3: if $\phi$ are LoH formulas it means we can substitute $\phi$ $[a,b]$ and hence the authors seem to imply that on the ground of how you write a formula you get different interpretations. How is that possible? I think the authors defer this choice to Appendix B - however, it remains unclear from the main text.

Regarding the experimental analysis:

1. It is not clear how many rules this method is extracting for each dataset and how this compares with the others method cited. For example how many rules does MLLP extract and how many does this method extract?

2. It would be very nice to include some examples of the obtained rules

3. On the tabular benchmarks it is not clear to me why one should use this method rather than the MLLP - what is the practical advantage?

4.  A big selling point of the method is that it can be stacked up on top of any neural network - why did the authors only perform experiments on such simple datasets? Did you try to use it on some dataset on which some constraints are known (e.g., CIFAR 100) and see how many you recover?

**Questions:**

See above

---

> ### Author Response · Authors · 2025-11-25
>
> **Regarding $\top$ and $\bot$:**
> Thanks for pointing out the problem. We do include $\top$ and $\bot$ in our framework: for instance, see formula (2) and (6). We omitted them from the formal definition of line 153 for brevity. Note that it is a common practice not to specify $\top$ and $\bot$ in the grammar definition: this convention is adopted for example in "A Mathematical Introduction to Logic" by Herbert B. Enderton, and in other standard textbooks in logic.
> In any case, we will add $\top$ and $\bot$ to LoH formal grammar to avoid any ambiguity.
>
> **Regarding $[]$:** As stated in line 154, we allow $n$ to vary among the _positive_ integers, so we avoid the pathological case $[]$. In fact, the choice operator is meant to select one formula, so we need at least one to choose from.
>
> **Regarding the concept of hypothesis space:** The hypothesis space is a well-known concept in machine learning: it is the set of all possible functions that a learning algorithm can choose from to map input data to output predictions. A feature of LoH is that the hypothesis space of a compiled LoH formula (as a machine learning model) corresponds to a set of propositional logic formulas (as Boolean functions), and these can be explicitly listed applying the procedure defined in Section 4.
>
> **Regarding R1-R5:** Thank your for pointing to a possible source of confusion between LoH formulas and the corresponding sets of propositional formulas. Hoping to make the distinction clearer, we rewrite rules R1-R5 in the following way:
> * $\mathcal{H}(v) = \\{v\\}$;  $\mathcal{H}(\top) = \\{\top\\}$; $\mathcal{H}(\bot) = \\{\bot\\}$;
> * $\mathcal{H}(\neg F_1) = \\{\neg \phi \mid \phi \in \mathcal{H}(F_1)\\}$;
> * $\mathcal{H}(F_1 \land F_2) = \\{\phi \land \psi \mid \phi \in \mathcal{H}(F_1), \psi \in \mathcal{H}(F_2)\\}$;
> * $\mathcal{H}(F_1 \lor F_2) = \\{\phi \lor \psi \mid \phi \in \mathcal{H}(F_1), \psi \in \mathcal{H}(F_2)\\}$;
> * $\mathcal{H}([F_1,\dots,F_n]) = \bigcup_{i=1}^n \mathcal{H}(F_i)$.
>
> These rules define $\mathcal{H}$ as an operator that takes a LoH formula and produces a set of propositional formulas. We also update Example 2 accordingly in the revised version.
>
> **Why introducing the choice operator rather than working directly in propositional logic.**
> LoH provides a modular and practical way to compactly define both the structure of the search space (the hypothesis space) and of the model itself.
> It is true that the LoH compiler generates a propositional formula. However, it does more:
> * It implements the new propositions (the weights) as learnable parameters (torch variables).
> * It implements operations to sample and add Gumble noise to those variables during training.
> * It includes the constraint that exactly one weight proposition is true for each choice operator, via the technique presented in the bullet points in page 5. Importantly, this constraint cannot be encoded directly in the propositional formula, because such formula represents a function transforming inputs and weights into outputs, and not a set of constraints to be satisfied.
>
> You also pointed out we sometimes treat the choice operator as conjunction and sometimes as disjunction. Notice that the difference is only on the optimization dynamics, since both compilations learn to select the single best subformula among the candidates.
>
> **Example 3:** The difference in interpretations arises from whether the choices are independent or shared.
> In $[a,b]\land[a,b]$ there are two distinct choice operators. Both can (independently) pick either $a$ or $b$, so
> four combinations $\\{a\land a,a\land b,b\land a,b\land b\\}$ are possible.
> If we use a placeholder $\phi:=[a,b]$ and write $\phi\land \phi$, we enforce that the same choice is made in both instances. Since there is a single shared choice, only the two combinations $\\{a\land a, b\land b\\}$ can occur.
> At the compiled model level, one way to think about it is that placeholders implement weight sharing: in $[a,b]\land[a,b]$, the weights of the two choice operators are different, while in $\phi\land\phi$ they are shared.
>
> **Regarding the extracted rules in the tabular experiments:** Tank you for the suggestion. We are working to add a table in camera-ready version comparing extracted rules for the different datasets and models.
>
> **Advantage over MLLP:** MLLP is a continuous model just like a standard neural network, and we argue that we should compare the accuracy of the extracted logical formulas.
> Our model provably keeps the same accuracy before and after discretization: our extracted rules are 100\% faithful to the model's performance (and inner working). This is not guaranteed by MLLP, and indeed the rules extracted from MLLP (which are called CRS) generally perform worse than both MLLP and LoH.

---

> > ### Comment · Reviewer_wHxv · 2025-11-27
> >
> > Thank you very much for your answers. There are still some points that need to be clarified.
> >
> > 1. While the general concept of hypothesis space is well known, the class of functions your model can learn very much depends on the algorithm used. Also, in this case the authors need to be able to write exactly which hypothesis they are considering as I do not think the hypothesis $e^x$ can be modelled.
> >
> > 2. In Example 3, how can I know where the choices are independent or shared? If the set $\mathcal{H}(\Phi)$ is obtained by applying inductively the operations R1-R5 how can we obtain different interpretations? Can you please give me all the steps?
> >
> > 3.  The authors have to replied to the question: "It is not clear how many rules this method is extracting for each dataset and how this compares with the others method cited. For example how many rules does MLLP extract and how many does this method extract?" I understand that you want to give a full table, but in the meantime can you provide some indicative numbers?

---

> > > ### Author Response · Authors · 2025-11-28
> > >
> > > We thank the reviewer for the questions and address each point below:
> > >
> > > 1. In LoH, the hypothesis space is determined by the LoH formula, which is specified by the model creator. It is a set of propositional formulas defining Boolean functions $f : \\{0,1\\}^n \to \\{0,1\\}$. Real-valued functions such as $x \mapsto e^x$ are outside of this scope. The procedure for computing the hypothesis space of a LoH formula is given by the inductive application of rules R1-R5 if no placeholder is used, and by algorithm 1 in the general case.
> > >
> > > 2. The distinction between shared and independent choices is determined syntactically by the presence or not of placeholder declarations. If no placeholder is declared, then the inductive rules R1-R5 allow to compute the hypothesis space. Otherwise, Appendix B is entirely devoted to the formal treatment of placeholders, and the main text explicitly refers the reader to this appendix immediately before Example 3.
> > > The complete procedure for enumerating the hypothesis space for LoH formulas possibly containing placeholders is reported as Algorithm 1.
> > >
> > > 3. We first note that the comparison is not fair: in these experiments, we let hyperparameter tuning choose the format of the formula (i.e. the sequence of logical layers such as `or(and)`, `and(or)`, `or(and(or(and)))`), both for CRS and for our method. For a fairer comparison, one would need to re-run all methods while fixing the same format across them; this is precisely what we plan to do in the camera-ready version.
> > >
> > >     With this caveat in mind, we now report explicit statistics in Table 1. For simplicity and to avoid ambiguities in how to simplify, count and aggregate clause lengths across multiple logical layers, we computed these statistics only for models with depth two (i.e., formats `or(and)`, `and(or)`). Before computing the numbers, we post-process the formulas produced by both CRS and our method by pruning dead nodes and removing redundant clauses. The reported values reflect the final, simplified rule sets, averaging across the multiple classes in the case of multiclass classification.
> > >
> > >     The observed clause lengths are strongly influenced by the chosen architecture: for instance, when the first layer is an `or`, the learned clauses may be longer because they combine multiple categories of the same one-hot encoded categorical variable into a single disjunctive group. Conversely, when the first layer is an `and`, each clause must draw from different categorical variables, which can yield shorter individual clauses.
> > >
> > >     Finally, we note that CRS exhibits a stronger tendency to use deeper architectures, whereas our method often favors shallower structures. From an interpretability perspective, increased depth can be even more problematic than increased clause length, as deeper nesting makes it harder to reason compositionally about the learned rules. Our approach tends to keep the depth modest, even at the cost of somewhat longer clauses in some datasets.
> > >
> > >
> > > **Table 1: Statistics of the extracted formulas.**
> > >
> > > | Dataset     | CRS: layers           | CRS: number clauses | CRS: clauses avg length | Ours: layers | Ours: number clauses     | Ours: clauses avg length   |
> > > |------------|----------------------|---------------|----------------------|-------------|--------------------|------------------------|
> > > | adult      | or(and(or(and)))     | -             | -                    | and(or)     | 10.7 ± 2.8         | 59.9 ± 9.0             |
> > > | bank-m.    | or(and(or(and)))     | -             | -                    | or(and(or)) | -                  | -                      |
> > > | banknote   | or(and)              | 8.4 ± 1.6     | 2.1 ± 0.2            | or(and)     | 6.1 ± 1.5          | 2.3 ± 0.3              |
> > > | blogger    | or(and)              | 5.5 ± 1.7     | 2.9 ± 0.2            | and(or)     | 6.0 ± 0.9          | 2.9 ± 0.1              |
> > > | chess      | or(and(or(and)))     | -             | -                    | and(or)     | 134.3 ± 29.8       | 18.5 ± 0.7             |
> > > | connect-4  | or(and)              | 68.3 ± 14.0   | 8.9 ± 1.4            | and(or)     | 2.2 ± 1.5          | 26.8 ± 14.0            |
> > > | letRecog   | or(and(or(and)))     | -             | -                    | and(or)     | 89.5 ± 8.9         | 55.9 ± 1.4             |
> > > | magic04    | or(and)              | 96.8 ± 9.1    | 3.9 ± 0.5            | and(or)     | 63.5 ± 13.8        | 37.8 ± 2.8             |
> > > | mushroom   | or(and)              | 6.3 ± 0.8     | 2.2 ± 0.6            | and(or)     | 4.3 ± 0.7          | 6.9 ± 0.2              |
> > > | nursery    | or(and(or(and)))     | -             | -                    | or(and(or)) | -                  | -                      |
> > > | tic-tac-toe| or(and)              | 9.9 ± 1.5     | 3.3 ± 0.3            | or(and)     | 11.6 ± 2.5         | 4.2 ± 0.4              |
> > > | wine       | or(and)              | 3.0 ± 1.6     | 3.7 ± 1.9            | and(or)     | 4.5 ± 1.7          | 9.7 ± 1.8              |

---

### Author Response · Authors · 2025-11-25
**Revised version uploaded**

We sincerely thank the reviewers for reading our manuscript and for the insightful and constructive feedback. We also apologize for our late reply during the rebuttal phase.

We have uploaded a revised version of the paper, in which all changes are highlighted in _red_ to make them easy to track. Briefly, the main modifications are the following:
* Motivated by reviewer wHxv’s feedback, we added $\top$ and $\bot$ to the formal grammar, and we rewrote rules R1-R5 and Example 2 to better separate LoH formulas and corresponding hypothesis spaces.
* We created the wildfire risk assessment task, to test LoH under full and partial knowledge. We present it in section 6 along with the results, and we provide full implementation details in Appendix E. In this way, we aim to strengthen the experimental analysis and explicitly cover the cases that reviewers Enp6 and UUDb correctly pointed out as not sufficiently covered in the original submission.

Page, line, and section numbers used in the rebuttal refer to the _original_ version of the manuscript.

---

### Meta-Review · Area_Chair_8kcD · 2025-12-27

**Summary:**

While the paper presents an interesting formalism for unifying knowledge injection and rule induction in a neurosymbolic setting, reviewers raised several substantial concerns that ultimately limit its suitability for acceptance. The novelty of the proposed choice operator and its conceptual positioning remain unclear, with multiple reviewers noting strong overlaps with existing differentiable logic, ILP, and probabilistic logic programming frameworks. The claimed unification and interpretability benefits are not convincingly demonstrated beyond what prior methods already provide.

Additionally, the experimental evaluation is considered limited in scope and scale. Most experiments focus on relatively simple or small-scale tasks, leaving questions about scalability, robustness, and effectiveness under partial or full prior knowledge largely unresolved. Several reviewers also found that interpretability claims were insufficiently supported by quantitative analysis or concrete comparisons. Taken together, these concerns about conceptual contribution, positioning relative to prior work, and experimental depth outweigh the paper’s strengths. Thus, this paper does not reach the acceptance standard.

**Reviewer Concerns:**

The rebuttal related to conceptual contribution may not address the reviewers' concerns.

**Reviewer Scores:**

The reviewers, especially the two reviewers who give 2 score, are expected to keep their original scores.

---

### Decision · Program_Chairs · 2026-01-26

Reject